# Stability, Complexity and Data-Dependent Worst-Case Generalization Bounds

## Abstract

Providing generalization guarantees for stochastic optimization algorithms remains a key challenge in learning theory. Recently, numerous works demonstrated the impact of the geometric properties of optimization trajectories on generalization performance. These works propose worst-case generalization bounds in terms of various notions of intrinsic dimension and/or topological complexity, which were found to empirically correlate with the generalization error. However, most of these approaches involve intractable mutual information terms, which limit a full understanding of the bounds. In contrast, some authors built on algorithmic stability to obtain worst-case bounds involving geometric quantities of a combinatorial nature, which are impractical to compute. In this paper, we address these limitations by combining empirically relevant complexity measures with a framework that avoids intractable quantities. To this end, we introduce the concept of *random set stability*, tailored for the data-dependent random sets produced by stochastic optimization algorithms. Within this framework, we show that the worst-case generalization error can be bounded in terms of (i) the random set stability parameter and (ii) empirically relevant, data- and algorithm-dependent complexity measures of the random set. Moreover, our framework improves existing topological generalization bounds by recovering previous complexity notions without relying on mutual information terms. Through a series of experiments in practically relevant settings, we validate our theory by evaluating the tightness of our bounds and the interplay between topological complexity and stability.

## 1 Introduction

Explaining the generalization capabilities of modern deep learning models is still an open problem and an active research area (Zhang et al., 2017; 2021). Classically, supervised learning tasks can be framed as a population risk minimization problem, defined as:

$$\min_{w \in \mathbb{R}^d} \left\{ \mathcal{R}(w) := \mathbb{E}_{z \sim \mu_z}[\ell(w, z)] \right\}, \tag{1}$$

where $z \in \mathcal{Z}$ denotes the data, following a probability distribution $\mu_z$ on the data space $\mathcal{Z}$, and $\ell : \mathbb{R}^d \times \mathcal{Z} \to \mathbb{R}$ is the loss function. In practice, we have $\mathcal{Z} = \mathcal{X} \times \mathcal{Y}$, where $\mathcal{X}$ denotes a feature space and $\mathcal{Y}$ is a label space. Since $\mu_z$ is unknown, it is in general not possible to directly solve Equation (1); instead, we typically solve the *empirical risk minimization problem*: for a dataset $S := (z_1, \ldots, z_n) \sim \mu_z^{\otimes n}$ sampled i.i.d. from $\mu_z$, it corresponds to

$$\min_{w \in \mathbb{R}^d} \left\{ \widehat{\mathcal{R}}_S(w) := \frac{1}{n} \sum_{i=1}^n \ell(w, z_i) \right\}. \tag{2}$$

To evaluate the validity of this replacement, it is essential to quantify the discrepancy between the two objectives. Thus, a central quantity of interest in learning theory is the *generalization error*, defined as $G_S(w) := \mathcal{R}(w) - \widehat{\mathcal{R}}_S(w)$.

To quantify the generalization error, a large body of work analyzes the problem at the level of a single weight vector $w$ by modeling the learning algorithm as a mapping $(S, U) \mapsto w_{S,U}$, where $S \in \mathcal{Z}^n$ represents the dataset and $U$ captures the algorithmic randomness. This viewpoint underlies, for example, information-theoretic approaches (Xu & Raginsky, 2017; Steinke & Zakynthinou, 2020; Pensia et al., 2018), as well as the algorithmic stability frameworks (Bousquet & Elisseeff, 2002; Hardt et al., 2016; Feldman & Vondrak, 2019). While these approaches provide generalization

guarantees in various settings, the choice of which iterate to analyze remains unclear in the absence of a stopping criterion, and consequently cannot capture the full behavior of the generalization error.

In contrast to focusing on a single weight vector, recent work has highlighted the role of the *parameter trajectory* – the sequence of iterates generated by the learning algorithm for solving Equation (2) – in explaining the generalization behavior of modern machine learning models (Xu et al., 2024; Fu et al., 2024; Andreeva et al., 2024). Indeed, since the empirical risk can contain many local minima, the trajectory surrounding a local minimum offers a concise way to assess its quality. The trajectory itself inherently reflects the influence of the optimization algorithm, the choice of hyperparameters, and the dataset, factors that are essential for deriving meaningful generalization bounds.

To capture the geometric information of the *parameter trajectory*, several authors have proposed analyzing the *worst-case* generalization error along the entire trajectory, rather than relying on a single weight vector. Due the dependence of the parameter trajectory on the random variables $S$ (dataset) and $U$ (algorithmic noise), numerous studies on worst-case generalization bounds adopted the formulation of *data-dependent random sets* (Simsekli et al., 2020; Dupuis et al., 2023; Andreeva et al., 2023). In line with these previous works, we formalize a learning algorithm as a mapping,

$$\mathcal{A} : \bigcup_{n \geq 1} \mathcal{Z}^n \times \Omega \longrightarrow \mathrm{CL}(\mathbb{R}^d) := \{\text{closed subsets in } \mathbb{R}^d\}, \quad \mathcal{A} : (S, U) \longmapsto \mathcal{W}_{S,U}, \quad (3)$$

where $S \in \mathcal{Z}^n$ is the dataset and $U \in \Omega$ is an independent random variable representing the algorithm randomness. This formalization includes (but is not limited to) the following examples.

**Example 1.1.** Consider a stochastic optimizer $w_{k+1} = F(w_k, S, U)$ ($U$ is the batch noise) and $K_1, K_2 \in \mathbb{N}^\star$. The parameter trajectory is defined as $\mathcal{A}(S, U) = \mathcal{W}_{S,U} := \{w_k, \ K_1 \leq k \leq K_2\}$.

**Example 1.2.** Consider a dynamics of the form $\mathrm{d}W_t = -\nabla \widehat{\mathcal{R}}_S(W_t)\mathrm{d}t + \sigma \mathrm{d}B_t$, where $U := (B_t)_{t \geq 0}$ is a Brownian motion. Let $T > 0$, the parameter trajectory is $\mathcal{W}_{S,U} := \{W_t, \ t \in [0, T]\}$.

The following Examples 1.3 and 1.4 illustrate that this formulation also includes classical learning-theoretic frameworks.

**Example 1.3.** Singleton bounds correspond to the case where we choose $\mathcal{A}(S, U) := \{w_{S,U}\}$.

**Example 1.4.** Classical worst-case (uniform) generalization error over a data-independent hypothesis set $\mathcal{W}$ (Shalev-Shwartz & Ben-David, 2014) correspond to a constant mapping $\mathcal{A}(S, U) = \mathcal{W}$.

In this general setting, the main quantity of interest is the *worst-case generalization error* over $\mathcal{W}_{S,U}$,

$$G_S(\mathcal{W}_{S,U}) := \sup_{w \in \mathcal{W}_{S,U}} \big( \mathcal{R}(w) - \widehat{\mathcal{R}}_S(w) \big). \quad (4)$$

In the case of Example 1.4, this problem has been classically analyzed through the Rademacher complexity over $\mathcal{W}$ (Bartlett & Mendelson, 2002). However, as noted by Dupuis et al. (2023), these classical techniques break down for data-dependent random sets.

To overcome this limitation, many studies have employed information-theoretic techniques, particularly within the "fractal-based" literature (Simsekli et al., 2020; Birdal et al., 2021). A unifying perspective recently emerged with the PAC-Bayesian theory for random sets (Dupuis et al., 2024), which was recently employed by Andreeva et al. (2024) to establish generalization bounds based on novel topological complexity measures. Informally, all these bounds are of the following form[1]:

$$\sup_{w \in \mathcal{W}_{S,U}} \big( \mathcal{R}(w) - \widehat{\mathcal{R}}_S(w) \big) \lesssim \sqrt{\frac{\mathbf{C}(\mathcal{W}_{S,U}) + \mathrm{IT} + \log(1/\zeta)}{n}}, \quad (5)$$

with probability at least $1 - \zeta$. The term IT is an *information-theoretic* (IT) term, typically the *total mutual information* between the dataset $S$ and the set $\mathcal{W}_{S,U}$ The aforementioned bounds differ in the choice of complexity measure $\mathbf{C}(\mathcal{W}_{S,U})$, but all include an IT term. For concrete examples, we refer the reader to (Dupuis et al., 2023; 2024; Andreeva et al., 2024). The presence of such IT term is a major drawback of these studies, as it is computationally intractable and not well-understood in the general case (Dupuis et al., 2024), and they can potentially be infinite.

On the other hand, Foster et al. (2019) adopted a conceptually different approach by extending the notion of algorithmic stability (Bousquet & Elisseeff, 2002) to data-dependent hypothesis sets. Despite allowing a different point of view on Equation (4), their assumption does not explicitly take into

---

[1]We use the notation $\lesssim$ in informal statements where absolute constants have been omitted.

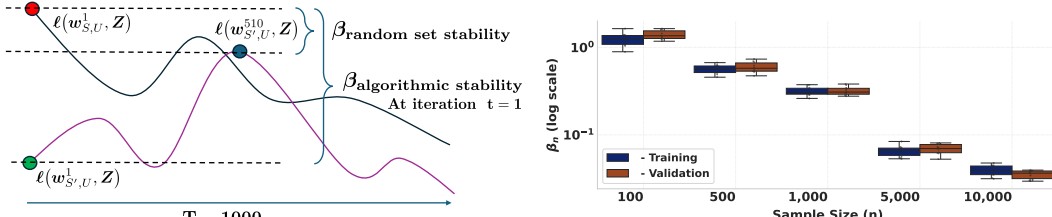

Figure 1: *(Left)* Evolution of the loss function $\ell(\cdot, Z)$ for a fixed sample $Z \in \mathcal{Z}$ over $T = 1000$ iterations, for two neighboring datasets $S, S' \in \mathcal{Z}^n$. While the classical notion of algorithmic stability measures the error at a specific iteration $t$, our stability notion extends this perspective to the entire training trajectory. *(Right)* Numerical estimation of our new random set stability parameter $(\beta_n)$ as $n$ increases. The experiments demonstrate that $\beta_n$ decreases with larger $n$.

account the algorithm randomness $U$, which is known to be paramount for deriving stability-based bounds (Hardt et al., 2016). Moreover, Foster et al. (2019) use a modified notion of Rademacher complexity whose evaluation requires all sets[2] of the form $\mathcal{W}_{S^\sigma}$, where $S_\sigma = (z_{1,\sigma_1}, \ldots, z_{n,\sigma_n})$, with $S_1 := (z_{1,1}, \ldots, z_{n,1}), S_{(-1)} := (z_{1,-1}, \ldots, z_{n,-1}) \sim \mu_z^{\otimes n}$ and $\sigma \in \{-1, 1\}^n$. Hence, the sets $\mathcal{W}_{S^\sigma}$ differ from the empirically accessible set $\mathcal{W}_{S,U}$ and evaluating all sets $\mathcal{W}_{S^\sigma}$ would require running the training algorithm exponentially many times in $n$, rendering this approach impractical. A similar construction has been used by Sachs et al. (2023) in the definition of the algorithmic-dependent Rademacher complexity.

**Contributions**. In this study, we propose to address the aforementioned issues by introducing a novel framework for deriving worst-case generalization bounds on data-dependent random sets, which are empirically relevant and do not contain intractable mutual information terms. Inspired by Foster et al. (2019), we adopt a stability-based approach, which we adapt to exploit the algorithm randomness in an explicit way. Based on our new assumption, we show that $G_S(\mathcal{W}_{S,U})$ can be upper bounded in terms of the stability parameter and a Rademacher complexity term. This allows us to take advantage of the best of both worlds by (1) avoiding the intricate information-theoretic (IT) terms and (2) relating the generalization error to topologically meaningful complexity measures, hence, providing IT-terms free versions of the topological generalization bounds of Birdal et al. (2021); Andreeva et al. (2024). Our contributions are detailed below.

- We introduce the notion of *random set stability*, specifically designed for stochastic learning algorithms. More precisely, we define a new stability parameter $\beta_n$ measuring how much the random set $\mathcal{W}_{S,U}$ changes if we replace one or more data points in $S$, in a sense that is formally introduced in Assumption 3.1 below. Moreover, we relate $\beta_n$ to classical stability notions, providing a systematic procedure to establish random set stability in diverse settings.

- We derive an expected worst-case generalization bound in terms of a Rademacher complexity term evaluated over the empirically relevant set $\mathcal{W}_{S,U}$ and the stability parameter.

- We apply our framework to obtain generalization bounds based on the fractal and topological complexity measures introduced in (Birdal et al., 2021; Andreeva et al., 2024) without any intractable mutual information terms. These bounds read as follows:

$$\mathbb{E}\left[\sup_{w \in \mathcal{W}_{S,U}} \left(\mathcal{R}(w) - \widehat{\mathcal{R}}_S(w)\right)\right] \lesssim \beta_n^{1/3}\left(1 + \mathbb{E}\left[\sqrt{\log \mathbf{C}(\mathcal{W}_{S,U})}\right]\right),$$

where $\beta_n$ is the stability parameter and $\mathbf{C}(\mathcal{W}_{S,U})$ is the complexity measure (see Section 4). Therefore, we provide the first fully computable topological bounds for practically used optimization algorithms, hence, addressing the main issues of the aforementioned previous works.

- We provide a systematic empirical investigation of the tightness of our bounds.

- We empirically investigate the previously introduced topological quantities and their interplay with stability as the sample size $n$ varies, extending the correlation analyses of prior studies, and therefore providing evidence supporting the validity of our stability-based approach.

The implementation will be made available upon publication. All proofs are in the appendix.

---

[2] $\mathcal{W}_S$ denotes in the setting of (Foster et al., 2019) a data dependent hypothesis set. The authors do not account for algorithmic randomness $U$. For $U$ constant both notation can be seen as equivalent.

## 2 TECHNICAL BACKGROUND

In this section, we provide some technical background on algorithmic stability. One of our contributions, described in Section 3.1, is to extend these notions to data-dependent random sets.

Algorithmic stability is a fundamental concept in learning theory (Bousquet & Elisseeff, 2002; Bousquet et al., 2020). It has proven successful in providing generalization guarantees for widely used stochastic optimization algorithms Feldman & Vondrak (2019); Zhu et al. (2023); Hardt et al. (2016). To position our work within existing stability-based bounds, we consider a variant of *algorithmic stability*, called *uniform argument stability* Bassily et al. (2020). As with most stability notions, it is defined with respect to a single iteration. Therefore, the learning algorithm used in the next definition corresponds to the setting of Example 1.3 and can be seen as a randomized mapping $(S, U) \longmapsto \mathcal{A}(S, U) \in \mathbb{R}^d$. In all the following, we say that two datasets $S, S' \in \mathcal{Z}^n$ are neighboring datasets if they differ by one element, which we denote $S \simeq S'$.

**Definition 2.1.** Let $\mathcal{A} : (S, U) \to \mathbb{R}^d$ be a randomized algorithm. $\mathcal{A}$ satisfies $\beta$-uniform argument stability if for any neighboring datasets $S \simeq S'$, we have $\mathbb{E}_U \left[ \|\mathcal{A}(S, U) - \mathcal{A}(S', U)\| \right] \leq \beta$.

If $\ell(w, z)$ is $L$-Lipschitz continuous in $w$, this implies the algorithmic stability Hardt et al. (2016):

$$\sup_{S \simeq S'} \sup_{z \in \mathcal{Z}} \mathbb{E}_U \left[ \ell(\mathcal{A}(S, U), z) - \ell(\mathcal{A}(S', U), z) \right] \leq L\beta. \tag{6}$$

In the context of worst-case generalization bounds, where $\mathcal{W}_{S,U}$ is a set (i.e., not a singleton), one needs to specify the stability of the entire set $\mathcal{W}_{S,U}$. To address this issue, a first step was taken by Foster et al. (2019), who introduced the following notion of hypothesis set stability.

**Definition 2.2.** A family $(\mathcal{W}_S)_{S \in \mathcal{Z}^n} \in (\mathbb{R}^d)^{\mathcal{Z}^n}$ of data-dependent hypothesis sets is $\beta$-uniformly stable when for any $S \simeq S'$ in $\mathcal{Z}^n$ and $w \in \mathcal{W}_S$, there exists $w' \in \mathcal{W}_{S'}$ such that:

$$\sup_{z \in \mathcal{Z}} |\ell(w, z) - \ell(w', z)| \leq \beta.$$

Although this assumption has been analyzed in settings such as strongly convex optimization (Foster et al., 2019), the notion of *hypothesis set stability* (Definition 2.2) does not account for the presence of algorithmic randomness $U$ and therefore cannot be applied to the data-dependent random sets defined by Equation (3). In Section 3, we thus extend Definition 2.2 to data-dependent random sets.

## 3 MAIN THEORETICAL RESULTS

In this section, we present our main theoretical contributions, which consist in deriving expected worst-case generalization bounds through a new concept of random set stability, described in Section 3.1. After discussing its validity, we present a key technical lemma and recover classical generalization bounds in the case of Examples 1.3 and 1.4. We then apply our framework to improve several data-dependent worst-case generalization bounds, including intrinsic dimension bounds Simsekli et al. (2020); Birdal et al. (2021) and topological bounds Andreeva et al. (2024).

### 3.1 RANDOM SET STABILITY

As explained above, the main drawback of Definition 2.2 is that it does not explicitly account for the algorithm randomness (e.g., batch indices), while algorithmic randomness is known to be paramount for single-iterates stability bounds Hardt et al. (2016); Zhu et al. (2023).

To address this issue, we propose an improvement of the assumption of Foster et al. (2019), accounting for the presence of randomness. The main difficulty that arises here is to give a meaning to the expressions such as "for all $w \in \mathcal{W}_{S,U}$", when $\mathcal{W}_{S,U}$ is a random set. To do so, we define the notion of data-dependent selection Molchanov (2017).

**Definition 3.1.** A data-dependent selection of $\mathcal{W}_{S,U}$ is a deterministic mapping $\omega : \mathrm{CL}(\mathbb{R}^d) \times \mathcal{Z}^n \to \mathbb{R}^d$ such that $\omega(\mathcal{W}_{S,U}, S') \in \mathcal{W}_{S,U}$, almost surely. In particular, we assume the existence of a random variable $\omega_0(\mathcal{W}_{S,U}, S')$ such that, almost surely, $\omega_0(\mathcal{W}_{S,U}, S') \in \arg\max_{w \in \mathcal{W}_{S,U}} G_{S'}(w)$, where we recall that $G_{S'}(w) := \mathcal{R}(w) - \widehat{\mathcal{R}}_{S'}(w)$ for all $S' \in \mathcal{Z}^n$.

Note that the existence of such a random variable is ensured under mild measure-theoretic conditions Molchanov (2017). Equipped with this definition, we now introduce our stability assumption.

**Assumption 3.1** (Random set stability). *Algorithm $\mathcal{A}$ is $\beta_n$-random set stable if for any $J \in \mathbb{N}^\star$ and any data-dependent selection $\omega$ of $\mathcal{W}_{S,U}$, there exists a map $\omega' : \mathrm{CL}(\mathbb{R}^d) \times \mathbb{R}^d \to \mathbb{R}^d$ such that:*

- *For any $S$, $U$ and $w \in \mathbb{R}^d$, $\omega'(\mathcal{W}_{S,U}, w) \in \mathcal{W}_{S,U}$ .*
- *For all $z \in \mathcal{Z}$ and two datasets $S, S' \in \mathcal{Z}^n$ differing by $J$ elements we have:*

$$\mathbb{E}_U[|\ell(\omega(\mathcal{W}_{S,U}, S), z) - \ell(\omega'(\mathcal{W}_{S',U}, \omega(\mathcal{W}_{S,U}, S)), z)|] \leq \beta_n J.$$

In the absence of algorithmic randomness (i.e., when $U$ is constant), Assumption 3.1 is a particular case of Definition 2.2. Stability assumptions are usually enounced for neighboring datasets; here, we present a variant with datasets differing by $J$ elements. These two variants are equivalent, and we choose this presentation to simplify subsequent proofs and notations. The presence of the mapping $\omega'$ might be intriguing: $\omega'$ corresponds to the parameter denoted $w'$ in Definition 2.2 and is denoted this way to make its dependence on the different random variables explicit.

**Applicability of random set stability**. The question naturally arises as to whether this assumption can be met and how it fits within the existing stability conditions. We answer this question with the following lemma, which shows that Assumption 3.1 is implied by Definition 2.1.

**Lemma 3.2.** *Consider a fixed $K \in \mathbb{N}^\star$, $1 \leq k \leq K$, and algorithms[3] $\mathcal{A}_k(S, U) := w_k$. Let $\mathcal{A}(S, U) = \mathcal{W}_{S,U} = \{\mathcal{A}_k(S, U)\}_{k=1}^K$ be an algorithm in the sense of Equation (3). Assume that for all $k$, $\mathcal{A}_k$ is $\delta_k$-uniformly argument-stable in the sense of Definition 2.1 and that the loss $\ell(w, z)$ is $L$-Lipschitz-continuous in $w$. Then, $\mathcal{A}$ is random set stable with parameter at most $\beta_n = L \sum_{k=1}^K \delta_k$.*

In the context of Example 1.1, this result shows that Definition 2.1 implies random set stability. It can be noted that many stability-based studies rely on a Lipschitz assumption and prove uniform argument stability to derive algorithmic stability parameters Hardt et al. (2016), as noted by Bassily et al. (2020). Therefore, we see that the conditions of Lemma 3.2 can be satisfied in numerous cases, such as optimization of convex, smooth, and Lipschitz continuous functions Hardt et al. (2016). Under these conditions, it is obtained that the stability parameter is of order $\delta_k = \mathcal{O}(k/n)$, hence, yielding random set stability with a parameter of order $\mathcal{O}(T^2/n)$, in the worst case.

**Projected SGD**. Fix $T \in \mathbb{N}$. Let us consider the projected SGD recursion on a loss function $\ell$, which can be non-convex, i.e., given a dataset $S = (z_1, \ldots, z_n) \in \mathcal{Z}^n$, consider the following algorithm:

$$w_{k+1} = \Pi_R \left[ w_k - \eta_k \nabla \ell(w_k, z_{i_{k+1}}) \right], \quad \mathcal{A}_{\mathrm{SGD}}(S, U) = \mathcal{W}_{S,U} := \{w_1, \ldots, w_T\} \tag{7}$$

where $i_k \sim \mathcal{U}(\{1, \ldots, n\})$ are random indices, $U := \{i_k\}_{k \geq 1}$, and $\Pi_R$ is the projection on the ball $\bar{B}_R(0)$. The next corollary uses Lemma 3.2 to establish the random set stability of SGD for Lipschitz and smooth losses, and is an adaptation of (Hardt et al., 2016, Theorem 3.12) to our setting.

**Corollary 3.3.** *Suppose that, for all $z$, $\ell(\cdot, z)$ is $L$-Lipschitz, $\nabla \ell(\cdot, z)$ is $G$-Lipschitz, and that $\eta_k \leq c/k$ for some $c < 1/G$. Then, $\mathcal{A}_{\mathrm{SGD}}$ is random-set-stable with parameter*

$$\beta_n = \frac{4LR}{n-1} \left( \frac{L}{GR} \right)^{1/Gc+1} \sum_{1 \leq k \leq T} k^{\frac{Gc}{Gc+1}}.$$

## 3.2 Worst-case generalization bound for data-dependent stable random sets

We now leverage Assumption 3.1 to prove worst-case generalization bounds over random sets. The next lemma shows that the expected worst-case generalization error can be bounded in terms of the random set stability parameter and a Rademacher complexity term (Bartlett & Mendelson, 2002).

**Lemma 3.4.** *Let $S \sim \mu_z^{\otimes n}$. Let $J, K \in \mathbb{N}^\star$ be such that $n = JK$, and $\tilde{S}_J \sim \mu_z^{\otimes J}$ be independent of $S$ and $U$. Assume that algorithm $\mathcal{A}$ satisfies Assumption 3.1 with parameter $\beta_n > 0$, then:*

$$\mathbb{E}\left[ \sup_{w \in \mathcal{W}_{S,U}} \left( \mathcal{R}(w) - \widehat{\mathcal{R}}_S(w) \right) \right] \leq 2\mathbb{E}\left[ \mathbf{Rad}_{\tilde{S}_J}(\mathcal{W}_{S,U}) \right] + 2J\beta_n, \tag{8}$$

*where $\mathbf{Rad}_S(\mathcal{W})$ denotes the Rademacher complexity, whose definition is recalled in Definition A.1.*

Most existing data-dependent worst-case generalization bounds rely on an information-theoretic approach Simsekli et al. (2020); Hodgkinson et al. (2022); Andreeva et al. (2024). In particular, this theorem can be seen as an analog of (Dupuis et al., 2024, Theorem 6), which provided a bound in

---

[3] $\mathcal{A}_k$ is an algorithm with values in $\mathbb{R}^d$, i.e., its output is the $k$-th iterate of the optimization algorithm.

terms of Rademacher complexity and information-theoretic terms. Instead of information-theoretic terms, our framework makes use of the random set stability parameter $\beta_n$.

In Lemma 3.4, the number $J$ is a free parameter. Before discussing the main applications of our methods, we show that a certain choice of $J$ recovers: (1) classical algorithmic stability bounds ($J = 1$) and (2) worst-case bounds over fixed hypothesis sets ($J = n$). Therefore, the parameter $J$ can be seen as interpolating between these two settings, while yielding new data-dependent worst-case generalization bounds for intermediary values of $J$.

**Corollary 3.5** (Recovering stability bounds). *Consider an algorithm $\mathcal{A}$ as in Example 1.3 and assume that $\mathcal{A}$ satisfies Assumption 3.1 with parameter $\beta_n$. Then, $\mathcal{A}$ is algorithmically stable in the sense of Equation* (6). *By setting $J = 1$ in Lemma 3.4, we obtain:*

$$\mathbb{E}\big[\mathcal{R}(\mathcal{A}(S, U)) - \widehat{\mathcal{R}}_S(\mathcal{A}(S, U))\big] \leq 2\beta_n.$$

Up to the absolute constant 2, this recovers classical algorithmic stability-based expected generalization bounds Bousquet & Elisseeff (2002); Bassily et al. (2020).

**Corollary 3.6** (Recovering bounds on fixed hypothesis sets). *Consider the setting of Example 1.4. Then, the algorithm satisfies Assumption 3.1 with parameter $\beta_n = 0$. By using $J = n$ we obtain:*

$$\mathbb{E}\big[\sup_{w \in \mathcal{W}} \big(\mathcal{R}(w) - \widehat{\mathcal{R}}_S(w)\big)\big] \leq 2\mathbb{E}\left[\mathbf{Rad}_S(\mathcal{W})\right],$$

*where we observe that $\tilde{S}_n \sim \mu_z^{\otimes n}$ and $S \sim \mu_z^{\otimes n}$ have the same distribution.*

We observe that when $\beta_n = \mathcal{O}\left(n^{-1}\right)$, Corollary 3.5 recovers the classical convergence rate of Hardt et al. (2016). Corollary 3.6 shows that our framework tightly recovers the classical Rademacher complexity bounds (for fixed hypothesis set $\mathcal{W}$) Bartlett & Mendelson (2002), as well as all known consequences of these bounds, such as VC dimension bounds (Vapnik & Vapnik, 1998) and covering number bounds (Shalev-Shwartz & Ben-David, 2014). In particluar, Corollary 3.6 shows that in the data-independent case we obtain the usual worst-case convergence rate of $\mathcal{O}\left(n^{-1/2}\right)$.

## 4 APPLICATIONS

In this section, we apply our random set stability framework to improve recently proposed data-dependent worst-case generalization bounds in terms of fractal dimensions and topological complexities Andreeva et al. (2024). We assume that a learning algorithm $\mathcal{A}(S, U) =: \mathcal{W}_{S,U}$ is given, in the sense of Equation (3). One may think of $\mathcal{W}_{S,U}$ as a learning trajectory as in Example 1.1.

**Assumptions**. Before presenting our main applications, we introduce some structural assumptions. We first note that the Rademacher complexity (RC) term appearing in Lemma 3.4 is computed using a sample $\tilde{S}_J \sim \mu_z^{\otimes J}$ independent of the random set $\mathcal{W}_{S,U}$. Despite this, the set over which the RC term is computed is the data-dependent random set $\mathcal{W}_{S,U}$, while the independent sample $\tilde{S}_J$ is only used to evaluate the loss inside the RC. In order to exploit this property, we assume that the loss has a "local" Lipschitz regularity, in the following sense.

**Assumption 4.1** (Lipschitz on random sets). *There exists a measurable map $(S, U) \mapsto L_{S,U}$ such that for all $S$ and all $U$, for all $w, w' \in \mathcal{W}_{S,U}$, for all $z \in \mathcal{Z}$, $|\ell(w, z) - \ell(w', z)| \leq L_{S,U}\|w - w'\|$.*

Lipschitz continuity assumptions are commonly used in the context data-dependent worst-case bounds Simsekli et al. (2020); Birdal et al. (2021); Andreeva et al. (2023). However, our framework crucially only requires the Lipschitz continuity of $\ell(\cdot, z)$ to hold on each set $\mathcal{W}_{S,U}$ individually, with a Lipschitz constant $L_{S,U}$ that depends both on the dataset $S \in \mathcal{Z}^n$ and the noise $U$. Note however that this local Lipschitz continuity of $\ell(\cdot, z)$ is still required to be uniform in $z \in \mathcal{Z}$. For instance, in the case of Example 1.1, where $\mathcal{W}_{S,U}$ is finite, then Assumption 4.1 is satisfied as soon as $\mathcal{Z}$ is finite (or compact if $z \mapsto \nabla\ell(w, z)$ is continuous for a topology on $\mathcal{Z}$).

As is typical in this literature Andreeva et al. (2024); Dupuis et al. (2024); Birdal et al. (2021); Bartlett & Mendelson (2002), we also assume the boundedness of the loss function.

**Assumption 4.2.** *We assume that the loss $\ell$ is bounded in $[0, B]$, with $B > 0$ being a constant.*

## 4.1 INTRINSIC DIMENSIONS AND TOPOLOGICAL BOUNDS

Recent studies Simsekli et al. (2020); Dupuis et al. (2023); Andreeva et al. (2023); Dupuis et al. (2024); Birdal et al. (2021); Andreeva et al. (2024) have established upper-bounds on Equation (4) and drawn links between the *topological structure* of $\mathcal{W}_{S,U}$ and the generalization error. These bounds are of the form presented in Equation (5). As mentioned in Section 1, one of the main drawbacks of these bounds is that they contain intractable and poorly understood mutual information terms. One of our main contributions is to show that, under random set stability, we are able to obtain data-dependent worst-case generalization bounds featuring many of the most popular complexity measures, without the need of the intractable mutual information terms (denoted IT in Equation (5)).

We first present an adaptation to our setting of the intrinsic dimension-based bounds of (Simsekli et al., 2020), without any mutual information term.

**Theorem 4.3.** *Suppose that Assumptions 4.2, 3.1, and 4.1 hold, and that $\mathcal{W}_{S,U}$ is a.s. of finite diameter. Without loss of generality, assume that $\beta_n^{-2/3}$ is an integer divisor of $n$. There exists $\delta_n > 0$ such that for all $\delta < \delta_n$*

$$\mathbb{E}\left[\sup_{w \in \mathcal{W}_{S,U}} \left(\mathcal{R}(w) - \widehat{\mathcal{R}}_S(w)\right)\right] \leq 2\mathbb{E}\left[\frac{B}{n} + \delta L_{S,U} + \beta_n^{1/3}\left(1 + B\sqrt{4\overline{\dim}_B(\mathcal{W}_{S,U})\log\frac{1}{\delta}}\right)\right],$$

*where $\overline{\dim}_B(\mathcal{W}_{S,U})$ is the upper box-counting dimension (Falconer, 2013), which we define in the appendix in Definition A.4. We refer to the Appendix B.4 for a discussion of the parameter $\delta_n$.*

In many cases, the upper box-counting dimension of $\mathcal{W}_{S,U}$ is much smaller than the ambient dimension $d$ (Falconer, 2013). Under mild assumptions, the upper box-counting dimension can be replaced by other notions like the persistent homology dimension (Birdal et al., 2021) and the magnitude dimension (Andreeva et al., 2023), which have provided promising empirical results.

However, it was argued by Andreeva et al. (2024) that such intrinsic dimensions are not well-suited to the practically used stochastic optimizers due to their discrete nature, as for instance in Example 1.1. To overcome this issue, these authors proposed to use two new notions of topological complexity, called the $\alpha$-weighted lifetime sums (denoted $\mathbf{E}^\alpha(\mathcal{W}_{S,U})$) and the positive magnitude (denoted $\mathbf{PMag}(s \cdot \mathcal{W}_{S,U})$), and significantly improved the empirical results. Intuitively, the $\alpha$-weighted lifetime can be understood as tracking the evolution of 'connected components' of a set across different scales and summing their contributions. On the other hand, the magnitude may be thought of as capturing the effective number of distinct points in a finite space at different scale Leinster (2013). We defer a precise definition of these notions to Definition A.6 and A.9.

In the next theorem, we provide IT free versions of the topological generalization bounds of Andreeva et al. (2024) under our assumption of random set stability (Assumption 3.1).

**Theorem 4.4.** *Suppose that Assumptions 4.2, 3.1, and 4.1 hold. We further assume that the set $\mathcal{W}_{S,U}$ is almost surely finite. Without loss of generality, assume that $\beta_n^{-2/3}$ is an integer divisor of $n$. Then, for any $\alpha \in (0, 1]$, we have*

$$\mathbb{E}\left[\sup_{w \in \mathcal{W}_{S,U}} (\mathcal{R}(w) - \widehat{\mathcal{R}}_S(w))\right] \leq \beta_n^{1/3}\left(2 + 2B + 2B\mathbb{E}\left[\sqrt{2\log(1 + K_{n,\alpha}\mathbf{E}^\alpha(\mathcal{W}_{S,U}))}\right]\right),$$

*where $K_{n,\alpha} := 2(2L_{S,U}\sqrt{n}/B)^\alpha$. Moreover, for any $\lambda > 0$, let $s(\lambda) := \lambda L_{S,U}\beta_n^{-1/3}/B$. We have*

$$\mathbb{E}\left[\sup_{w \in \mathcal{W}_{S,U}} (\mathcal{R}(w) - \widehat{\mathcal{R}}_S(w))\right] \leq \beta_n^{1/3}\left(2 + \lambda B + \frac{2B}{\lambda}\mathbb{E}\left[\log\mathbf{PMag}\left(s(\lambda) \cdot \mathcal{W}_{S,U}\right)\right]\right).$$

This result suggests that a theoretically justified choice of the magnitude scale would be $s(\lambda) \approx \beta_n^{-1/3}$, which amounts to $\Theta(n^{1/3})$ in the event that $\beta_n = \mathcal{O}(1/n)$. As a reminder, in the classical setting Corollary 3.6 we recover the standard convergence rate of $\mathcal{O}(n^{-1/2})$.

Unlike trajectory-dependent topological bounds Simsekli et al. (2020); Birdal et al. (2021); Dupuis et al. (2023); Andreeva et al. (2024), our bounds do not involve IT terms, which can be unbounded. While our bounds result in a slower convergence rate, this represents a deliberate trade-off to maintain boundedness. In addition to replacing IT terms with a stability assumption, the stability parameter $\beta_n$ acts as a multiplying constant in the bound, hence increasing the impact of the topological

complexity on the bound. Compared to the IT terms, $\beta_n$ is easily interpretable and can be expected to decrease with $n$ at a polynomial rate, as discussed above.

## 5 EMPIRICAL EVALUATIONS

In Section 4, we provided the first fully computable topological/worst-case generalization bounds, without the need for intractable mutual information terms, which limited the empirical analysis of previous works. Instead, our bounds are mainly based on the stability parameter $\beta_n$, which allows us to assess how they empirically compare with the worst-case generalization error and whether their structure aligns with observations, thus validating the framework and its practical applicability.

All experiments are performed using a Vision Transformer (ViT) Dosovitskiy et al. (2021) on the CIFAR-100 dataset and GraphSAGE Hamilton et al. (2017) on the MNISTSuperpixels dataset Monti et al. (2017). All models are implemented in PyTorch and trained with the ADAM optimizer (Kingma & Ba, 2014). Further implementation details for both models are provided in Appendix C.

**Experimental Design**. We follow the protocol of previous works Dupuis et al. (2023); Andreeva et al. (2024) and train models until convergence, using this checkpoint as a starting point for further analysis. Our empirical analysis is structured in two main parts.

First, we analyze the order of magnitude of our bounds, offering initial insights into the tightness of our framework. We fine-tune each model for 500 iterations on a previously unseen dataset, with fixed batch sizes ($b$) and learning rates ($\eta$), in particular $b \in \{64, 128\}$ and $\eta \in \{10^{-6}, 10^{-5}\}$. For each ($b, \eta$) configuration, we estimate the stability coefficient Assumption 3.1 over 5 different seeds.

Then we examine the relationship between stability and topological complexity (i.e., weighted lifetime sums and positive magnitude in Theorem 4.4). This analysis is motivated by the dependence of the stability parameter on the sample size $n$ and the multiplicative interaction between stability and topological complexity in Theorem 4.4. To this end, we fine-tune the models for further $5 \cdot 10^3$ iterations. The experiments are structured on a $4 \times 4 \times 5$ grid based on the learning rate $\eta$, the batch size $b$, and $n$. For detailed information about the hyperparameter, we refer to Appendix C.

**Quantity Estimation**. In this paragraph, we describe the quantities used in our empirical analysis and their estimation. Implementation details are provided in the Appendix.

- **Generalization error:** We approximate $G_S(\mathcal{W}_{S,U})$ (Equation (4)) by tracking train and test risk at every iteration and reporting $\max_t\{\texttt{test risk} - \texttt{train risk}\}$. This refines prior proxies based on either the final gap Birdal et al. (2021) or the worst test risk Andreeva et al. (2024).
- **Topological complexities:** We compute $\mathbf{PMag}(\mathcal{W}_{S,U})$ via a Krylov subspace PCG approximation Salim (2021), and $\mathbf{E}^\alpha(\mathcal{W}_{S,U})$ using $\texttt{giotto-ph}$ Pérez et al. (2021) with $\alpha = 1$.
- **Distance matrix:** For trajectories $\mathcal{W}_{S,U}$, we form a Euclidean distance matrix $\rho(w, w') = \|w - w'\|$. To reduce computational cost, we uniformly sample 1500 of the 5000 iterations.
- **Stability parameter:** For random set stability (Assumption 3.1), we replace 50 unseen samples per training set, retrain to obtain $\mathcal{W}_{S,U}$ and $\mathcal{W}_{S',U}$, and measure worst-case loss deviations across iterations $\max_{w \in \mathcal{W}_{S,U}} \min_{w' \in \mathcal{W}_{S',U}} \sup_{Z \in \mathcal{Z}} |\ell(w, Z) - \ell(w', Z)|$. With $M = 500$ held-out points $\mathbf{Z}$ (see Algorithm 1), we average results over 5 different random seeds. Note that this method necessarily leads to an optimistic estimation of the stability parameter $\beta_n$, as it would be intractable to evaluate the supremum over the entire data space $\mathcal{Z}$.

### 5.1 RESULTS AND DISCUSSION

We now present our main empirical results. First, we numerically evaluate our bounds. Next, we investigate the interplay between stability and the topological quantities appearing in Theorem 4.4.

**Order of the bounds**. Our goal is to have an estimate of the order of magnitude of the worst-case error predicted by our bounds. To avoid the computationally costly evaluation of Lipschitz constants, we estimate a simple upper bound on the Rademacher complexity that is common to all our theoretical results. Concretely, we use Massart's lemma (Shalev-Shwartz & Ben-David, 2014) to bound the right-hand side of Equation (8) by $2\sqrt{2\log(T)/J} + 2J\beta_n$, where $T$ is the number of iterations. We estimate $\beta_n$ using the procedure above and optimize over $J$ (see Appendix C.3).

We present the resulting estimation of our bound in Table 1, for different learning rates and batch sizes. A consistent pattern can be observed: the estimated bounds are typically close to an order

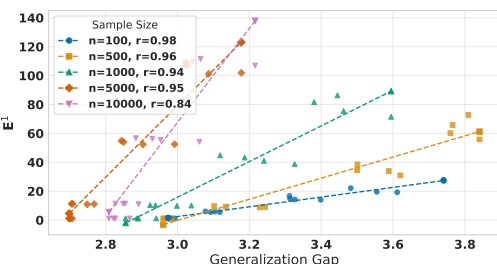 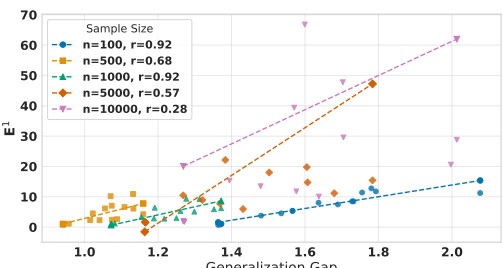

Figure 2: Variation of $\mathbf{E}^1$ with sample size $n$ for the model **ViT**. Pearson correlation coefficients $r$ are reported for each subgroup.

Figure 3: Variation of $\mathbf{E}^1$ with sample size $n$ for the model **GraphSage**. Pearson correlation coefficients $r$ are reported for each subgroup.

of magnitude larger than the actual worst-case generalization error. However, in most experimental settings, the estimated bounds remain below $100\%$ accuracy, hence, provide meaningful guarantees. We also find that the stability parameter $\beta_n$ varies with the hyperparameters $(\eta, b)$ and, therefore, encodes the impact of the learning algorithm on generalization error. Moreover, the estimated $\beta_n$ captures the generalization behavior: specifically, $G_S(\mathcal{W}_{S,U})$ tends to decrease as $\beta_n$ becomes smaller, a trend that holds for both ViT and GraphSage.

To the best of our knowledge, we are the first to *fully* estimate a bound on the worst-case error, thereby providing a stronger guarantee than in prior work. In contrast, previous studies focused on bounds for a single weight vector and similarly reported discrepancies of one to two orders of magnitude between the estimated bound and the actual error Chen et al. (2018); Li et al. (2019); Dupuis et al. (2023); Harel et al. (2025). This sug-

Table 1: Comparison of ViT and GraphSage across different learning rates ($\eta$) and batch sizes ($b$). We report $\beta_n$, $G_S(W_{S,U})$, and the resulting bound. We use the 0-1 loss.

| Model | $\eta$ | $b$ | $10^4 \cdot \beta_n$ | $10^2 \cdot G_S(W_{S,U})$ | $10^2 \cdot$ **Bound** |
|---|---|---|---|---|---|
| ViT | $10^{-4}$ | 64 | $4.72 \pm 0.52$ | $10.24 \pm 0.36$ | 104.43 |
| | $10^{-4}$ | 128 | $5.52 \pm 0.67$ | $12.84 \pm 0.55$ | 105.24 |
| | $10^{-5}$ | 64 | $2.16 \pm 0.36$ | $7.16 \pm 0.33$ | 68.47 |
| | $10^{-5}$ | 128 | $2.24 \pm 0.22$ | $6.36 \pm 0.33$ | 68.67 |
| GraphSage | $10^{-4}$ | 64 | $2.56 \pm 0.36$ | $5.48 \pm 0.36$ | 75.63 |
| | $10^{-4}$ | 128 | $2.64 \pm 0.22$ | $5.20 \pm 0.40$ | 75.79 |
| | $10^{-5}$ | 64 | $0.64 \pm 0.22$ | $4.60 \pm 0.14$ | 47.79 |
| | $10^{-5}$ | 128 | $0.80 \pm 0.00$ | $4.84 \pm 0.17$ | 48.59 |

gests that our worst-case generalization bounds are reasonable tight. Moreover, the fact that smaller estimated generalization gaps are consistently associated with smaller stability parameters and, therefore, smaller bounds, shows that the bounds adapt meaningfully to model performance.

**Interplay between Stability and** $\mathbf{C}(\mathcal{W}_{S,U})$. In Figure 2 and 3, we observe that the sensitivity of $\mathbf{E}^1(\mathcal{W}_{S,U})$ with respect to the generalization gap (i.e., the slope of the affine regression of $\mathbf{E}^1(\mathcal{W}_{S,U})$ by the value of $G_S(\mathcal{W}_{S,U})$) increases when $n$ gets larger, which is what is predicted by our theory. Indeed, Theorem 4.4 assert that $\log \mathbf{E}^1(\mathcal{W}_{S,U})$ should be (approximately) of order at least $\beta_n^{-1/3} G_S(\mathcal{W}_{S,U})$, which amounts to $n^{1/3} G_S(\mathcal{W}_{S,U})$ in the event that $\beta_n = \Theta(1/n)$. Therefore, our experimental results strongly support Theorem 4.4. Together, these observations unveil a strong coupling between stability and the topological complexity of the training trajectories. Moreover, the topological complexities increase for large $n$ and therefore become more relevant to the bound. The empirically observed coupling is supported by the theoretical results of Section 3, through the product of the stability parameter $\beta_n$ and the topological complexities. Finally, we observe that the reported Pearson correlations are lower for bigger $n$, especially for the GraphSage experiment. We believe that it could be due to the fact that reaching local minima is harder when $n$ increases, which can decrease the correlation between the topological complexities and the generalization error, as it was already reported in (Birdal et al., 2021; Andreeva et al., 2024).

## 6 CONCLUDING REMARKS

In this work, we present a new framework for obtaining worst-case generalization bounds over data-dependent random sets. This framework is based on a new concept of *random set stability*, which we demonstrate holds for practically used algorithms, and is related to classical stability notions. We then bound the worst-case generalization error by the (weighted) sum of the random set stability parameter and a Rademacher complexity term. We present several applications of our new techniques by recovering classical generalization bounds and proving mutual information-free versions

of existing fractal and topological bounds, which makes them for the first time fully computable. We supported our theory through a series of experiments conducted in practically relevant settings.

**Limitations & future work**. Despite providing the first mutual information-free topological and fractal bounds, our main results have two main disadvantages: *(i)* we only provide expected bounds, and *(ii)* our framework only allows for Euclidean-based topological complexities, and we typically do not cover the case of the data-dependent pseudometric introduced by Dupuis et al. (2023). Other methods to remove information-theoretic terms exist in the literature. For instance, Neu et al. (2021) added Gaussian noise to the algorithm output in their study of the generalization error of SGD. Extending such a technique to the random set setting is a promising direction for future works. Finally, while our paper extends uniform stability to random sets, we believe that other notions of stability, like the one of Lei & Ying (2020) could be extended to our random set setting and might lead to better generalization bounds under specific assumptions.

**Ethics statement**. This work adheres to the ICLR Code of Ethics, ensuring responsible use of publicly available datasets and maintaining full transparency in experimental design and reporting. The study complies with ethical guidelines on the integrity of research and considers possible social impacts. We have not used data collected on human subjects. In order to minimize carbon emissions, we utilized high-end GPUs only when they are necessary to train our large models. Since contributions are primarily theoretical, the focus is on advancing our understanding of the theory of artificial intelligence rather than targeting specific applications that may have direct societal impacts.

**Reproducibility Statement**. All data sets used in this work are referenced and publicly accessible. The experimental configurations and computational environment are outlined in detail within the main text and the appendix. The original code, accompanied by instructions, is included in the Supplementary Material. Lastly, all random seeds are provided to facilitate precise reproducibility. We plan to make our implementation available upon publication.

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

APPENDICES

We now provide additional technical details and proofs that are omitted from the main write-up. Further, we provide additional technical and empirical results. The appendix is thus organized in the following way:

- **Appendix A** provides supplementary technical background.
- **Appendix B** contains the omitted proofs of all our theoretical results.
- **Appendix C** details the experimental setup and procedures required for reproducibility.
- **Appendix D** presents additional empirical results.

# A   ADDITIONAL TECHNICAL BACKGROUND

In this section, we provide the relevant definitions as well as the technical background for our results.

## A.1   RADEMACHER COMPLEXITY

Recent advances on topological generalization bounds are based on a data-dependent Rademacher complexity, leveraging PAC-Bayesian theory on random sets Dupuis et al. (2024). In our stability-based approach, the Rademacher complexity also plays a central role. We define the standard Rademacher complexity Shalev-Shwartz & Ben-David (2014); Bartlett & Mendelson (2002) as:

**Definition A.1.** Let $\mathcal{W} \subset \mathbb{R}^d$, $S \in \mathcal{Z}^n$, and $\boldsymbol{\epsilon} := (\epsilon_1, \ldots, \epsilon_n) \sim \mathcal{U}(\{-1, 1\})^{\otimes n}$ be i.i.d Rademacher random variables (i.e., centered Bernoulli variables). Let $\ell : \mathcal{W} \times \mathcal{Z} \to \mathbb{R}_+$ be a loss function. We define the Rademacher complexity of $\mathcal{W}$ (relative to $\ell$) as:

$$\mathbf{Rad}_S(\mathcal{W}) := \mathbb{E}_{\boldsymbol{\epsilon}} \left[ \sup_{w \in \mathcal{W}} \frac{1}{n} \sum_{i=1}^n \epsilon_i \ell(w, z_i) \right].$$

Intuitively, the Rademacher complexity measures the ability of a model to fit random labels. Using this definition, we now establish our main result.

## A.2   COVERING AND PACKING NUMBERS

In this section, we provide definitions of covering and packing numbers. These quantities have long been central in learning theory, particularly in the context of classical covering-based arguments for Rademacher complexity Shalev-Shwartz & Ben-David (2014); Bartlett & Mendelson (2002).

**Definition A.2** (Covering number). Let $(\mathcal{W}, \rho)$ be a metric space with $\mathcal{W} \subset \mathbb{R}^d$ of finite diameter, and let $\delta > 0$. The *covering number* $N_\delta(\mathcal{W})$ is the smallest integer $K \in \mathbb{N}^\star$ such that there exist points $x_1, \ldots, x_K \in \mathcal{W}$ satisfying

$$\mathcal{W} \subset \bigcup_{k=1}^K B_\rho(x_k, \delta),$$

where $B_\rho(x, \delta)$ denotes the open ball of radius $\delta$ centered at $x$ with respect to the metric $\rho$.

In this definition, we take the centers $(x_1, \ldots, x_K)$ of the covering to be in $\mathcal{W}$. This allows us to leverage the "local" Lipschitz continuity assumption defined in Assumption 4.1. This definition does not lose any generality compared to taking the points in $\mathbb{R}^d$. In particular, both definitions yield the same fractal dimensions (Falconer, 2013).

**Definition A.3** (Packing number). Let $(\mathcal{W}, \rho)$ be a metric space and $\delta > 0$. The *packing number* $P_\delta^\rho(\mathcal{W})$ is the cardinality of a maximal set of points in $\mathcal{W}$ such that the closed balls of radius $\delta$ centered at these points are pairwise disjoint.

## A.3 INTRINSIC DIMENSIONS

In this section, we focus on the upper box-counting dimension (also known as the upper Minkowski dimension), one of the most widely used notions of fractal dimension Falconer (2013); Mattila (1999). This notion has also found applications in machine learning, for instance in the works of Simsekli et al. (2020); Dupuis et al. (2023).

**Definition A.4** (Upper box-counting dimension). Let $(\mathcal{W}, \rho)$ be a metric space of finite diameter and let $(|N_\delta(\mathcal{W})|)_{\delta > 0}$ denote the covering numbers introduced in Definition A.2. The upper box-counting dimension is defined by:

$$\overline{\dim}_{\mathrm{B}}(\mathcal{W}) := \limsup_{\delta \to 0} \frac{\log |N_\delta(\mathcal{W})|}{\log(1/\delta)}.$$

We refer the reader to (Falconer, 2013) for more technical background on this notion.

**Remark A.1.** It has been shown in Kozma et al. (2006); Schweinhart (2020) that, for any compact metric space, several notions of dimension coincide with the well-known upper box-counting dimension (see Definition A.4). In particular, the celebrated *persistent homology dimension* Adams et al. (2020) has been connected to generalization Birdal et al. (2021) and has also been studied empirically in various works Andreeva et al. (2024); Tan et al. (2024).

## A.4 TOPOLOGICAL COMPLEXITIES

In this section, we define the topological quantities appearing in Theorem 4.4.

$\alpha$**-weighted lifetime sum ($\mathbf{E}^\alpha$).** The weighted lifetime sums is a central notion in the study of persistent homology, a subfield of topological data analysis (TDA). The interested reader may find a full exposition of these notions in (Boissonnat et al., 2018; Schweinhart, 2020; Kozma et al., 2006). There exist several equivalent definitions the weighted lifetime sums of degree 0, see (Andreeva et al., 2024) for a concise review of these definitions in the context of machine learning.

In this section, we present an intuitive approach to the weighted lifetime sum, which we also restrict to finite subsets of $\mathbb{R}^d$ and to homology of dimension 0. Note that this is a simplified, non-standard (though equivalent) definition of $\mathrm{PH}^0$.

**Definition A.5** (Persistent homology of degree 0 $\mathrm{PH}^0$). Let $\mathcal{W} \subset \mathbb{R}^d$ be a finite set with cardinality $N$, endowed with the Euclidean metric. For each $t \geq 0$ (which is often referred to as a "time" variable in the TDA literature), we construct an undirected graph $G_t$ with edges given by

$$\forall x, y \in \mathcal{W}, \quad \{x, y\} \in G_t \iff \rho(x, y) \leq t.$$

There exists a finite sequence of times $0 < t_1 < \cdots < t_k < +\infty$ such that the number of connected components in $G_{t_i}$ changes compared to $G_t$ for $t < t_i$. Let $c_i$ denote the number of connected components in $G_{t_i}$. By convention, set $c_0 = N$ and $t_0 = 0$, and define $n_i := c_i - c_{i-1}$. Then $\mathrm{PH}^0$ is defined as the multi-set,

$$\mathrm{PH}^0(\mathcal{W}) := \Big\{\!\!\Big\{ \underbrace{t_1, \ldots, t_1}_{n_1 \text{ times}}, t_2 \ldots, \underbrace{t_k, \ldots, t_k}_{n_k \text{ times}} \Big\}\!\!\Big\},$$

where the notation $\{\!\{\cdot\}\!\}$ denotes a multi-set.

**Remark A.2.** The construction of the graph above consists in looking at the 1-dimensional simplices of the Rips complex associated with $\mathcal{W}$, see (Boissonnat et al., 2018, Chapter 2).

We now use $\mathrm{PH}^0$ to define the key quantities in our work, in particular the $\alpha$-weighted sum $\mathbf{E}^\alpha_\rho$.

**Definition A.6** ($\alpha$-weighted lifetime sums). Let $\mathcal{W} \subset \mathbb{R}^d$ be a finite subsets of $\mathbb{R}^d$, endowed with the Euclidean metric. For $\alpha \geq 0$, the $\alpha$-*weighted lifetime sum* of $\mathcal{W}$ is defined as

$$\mathbf{E}^\alpha(\mathcal{W}) := \sum_{t \in \mathrm{PH}^0(\mathcal{W})} t^\alpha,$$

where $\mathrm{PH}^0$ denotes the set of lifetimes of the 0-dimensional persistent homology intervals of $\mathcal{W}$, or, equivalently, the above definition.

**Magnitude and positive magnitude**. Magnitude is another known topological invariant used in TDA, it was introduced by Leinster (2013). In the context of machine learning, it has been used in particular by Andreeva et al. (2023) to analyze the generalization error of neural networks.

A detailed account of this theory can be found in (Meckes, 2013; 2015). In this paragraph, we restrict ourselves to a particular case, that is enough for our purpose. In particular, we only define these notions for finite sets, but all our theoretical results can be seamlessly extended to compact subsets of $\mathbb{R}^d$ (see (Andreeva et al., 2024, Remark B.15)).

In all this paragraph, we fix a finite set $\mathcal{W} \subset \mathbb{R}^d$. The key notion behind both magnitude and positive magnitude is that of a weighting of $\mathcal{W}$, which is defined below.

**Definition A.7** (Weighting). Let $\lambda > 0$. A $\lambda$-weighting of $\mathcal{W}$ is a vector $\beta_\lambda \in \mathbb{R}^{\mathcal{W}}$ such that:

$$\forall a \in \mathcal{W}, \ \sum_{b \in \mathcal{W}} e^{-\lambda \|a-b\|} \beta_\lambda(b) = 1.$$

The existence of such a weighting has been shown by Leinster (2013).

We can now define the magnitude of $\mathcal{W}$.

**Definition A.8** (Magnitude, (Leinster, 2013)). With the notations of Definition A.7, the magnitude function of $\mathcal{W}$ is defined as:

$$(0, +\infty) \ni \lambda \longmapsto \mathbf{Mag}(\lambda \cdot \mathcal{W}) := \sum_{a \in \mathcal{W}} \beta_\lambda(a).$$

Recently, Andreeva et al. (2024) demonstrated that by slightly adapting the definition of magnitude, one obtains an object that is naturally related to the Rademacher complexity (Definition A.1), hence, relating it to learning theory. Therefore, they introduced the notion of *positive magnitude*, which we recall below.

**Definition A.9** (Positive magnitude, (Andreeva et al., 2024)). With the notations of Definition A.7, the *positive* magnitude function of $\mathcal{W}$ is defined as:

$$(0, +\infty) \ni \lambda \longmapsto \mathbf{PMag}(\lambda \cdot \mathcal{W}) := \sum_{a \in \mathcal{W}} \left(\beta_\lambda\right)_+ (a),$$

where $x_+ := \max(x, 0)$.

# B  OMITTED PROOFS

In this section, we present the omitted proofs in full detail. We also present a few additional technical results that are useful for our proofs and may be of independent interest.

## B.1  PROOF OF COROLLARY 3.3

We give below the proof of Corollary 3.3, which provides a random set stability assumption for projected SGD, in smooth and Lipschitz continuous settings.

**Corollary 3.3.** *Suppose that, for all $z$, $\ell(\cdot, z)$ is $L$-Lipschitz, $\nabla\ell(\cdot, z)$ is $G$-Lipschitz, and that $\eta_k \leq c/k$ for some $c < 1/G$. Then, $\mathcal{A}_{\text{SGD}}$ is random-set-stable with parameter*

$$\beta_n = \frac{4LR}{n-1} \left(\frac{L}{GR}\right)^{1/Gc+1} \sum_{1 \leq k \leq T} k^{\frac{Gc}{Gc+1}}.$$

*Proof.* To show the trajectory-stability of SGD, we will first show that it is argument-stable in the sense of Definition 2.1, then we will use Lemma 3.2 to establish random set stability.

To prove the argument stability, we almost exactly follow the steps in (Hardt et al., 2016, Theorem 3.12). Let us consider a neighboring dataset $S' = (z'_1, \ldots, z'_n) \in \mathcal{Z}^n$ that differs from $S$ by one element, i.e., $z_i = z'_i$ except one element. Consider the SGD recursion on $S'$:

$$w'_{k+1} = \Pi_R \left[ w'_k - \eta_k \nabla\ell(w'_k, z'_{i_{k+1}}) \right].$$

Since most of $S$ and $S'$ coincide, at the first iterations it is highly likely that $w_k$ and $w'_k$ will be identical, since they will pick the same data points. Let us denote the random variable $\kappa \geq 1$ that is the first time $z_{i_\kappa} \neq z'_{i_\kappa}$. Consider $K \in \mathbb{N}$ with $1 \leq K \leq T$. By using this construction, we have:

$$\mathbb{E}_U[\|w_K - w'_K\|] = \mathbb{E}_U\left[\|w_K - w'_K\| \, \mathbb{1}_{\kappa \leq t_0}\right] + \mathbb{E}_U\left[\|w_K - w'_K\| \, \mathbb{1}_{\kappa > t_0}\right].$$

We now focus on the first term in the right-hand-side of the above equation:

$$\begin{aligned}
\mathbb{E}_U\left[\|w_K - w'_K\|\right] &\leq \mathbb{E}_U\left[(\|w_K\| + \|w'_K\|) \, \mathbb{1}_{\kappa \leq t_0}\right] \\
&\leq 2R \, \mathbb{P}(\kappa \leq t_0) \\
&= \frac{2Rt_0}{n}.
\end{aligned} \tag{9}$$

The second parameter is upper-bounded in the proof of (Hardt et al., 2016, Theorem 3.12) (adapted to our setting by noting that the projection on the ball is 1-Lipschitz), which reads as follows:

$$\mathbb{E}_U\left[\|w_K - w'_K\| \, \mathbb{1}_{\kappa > t_0}\right] \leq \frac{2L}{G(n-1)} \left(\frac{K}{t_0}\right)^{Gc}.$$

By combining these two estimates, we obtain:

$$\mathbb{E}_U[\|w_K - w'_K\|] \leq \frac{2Rt_0}{n} + \frac{2L}{G(n-1)} \left(\frac{K}{t_0}\right)^{Gc}.$$

Choosing

$$t_0 = \left(\frac{L}{GR}\right)^{\frac{1}{Gc+1}} K^{\frac{Gc}{Gc+1}}$$

yields

$$\begin{aligned}
\mathbb{E}_U[\|w_K - w'_K\|] &\leq \frac{4R}{n-1} \left(\frac{L}{GR}\right)^{\frac{1}{Gc+1}} K^{\frac{Gc}{Gc+1}} \\
&=: \delta_K.
\end{aligned}$$

Since $\delta_K$ is independent of $S$ and $S'$, this shows that the $k$-th iterate of SGD is $\delta_k$ argument-stable. Finally, we conclude by Lemma 3.2, that SGD is trajectory-stable with parameter

$$\beta_n = L \sum_{k=1}^{K} \delta_k.$$

This concludes the proof. $\qquad\square$

## B.2 A KEY TECHNICAL LEMMA: RADEMACHER BOUND FOR TRAJECTORY-STABLE ALGORITHMS

A key technical element of our approach is the following theorem, which presents an expected Rademacher complexity bound for the worst-case generalization error of trajectory-stable algorithms. Before proceeding to the proof, we recall the following notation for the worst-case generalization error.

$$G_S(\mathcal{W}) := \sup_{w \in \mathcal{W}} \left(\mathcal{R}(w) - \widehat{\mathcal{R}}_S(w)\right).$$

**Lemma 3.4.** *Let $S \sim \mu_z^{\otimes n}$. Let $J, K \in \mathbb{N}^\star$ be such that $n = JK$, and $\tilde{S}_J \sim \mu_z^{\otimes J}$ be independent of $S$ and $U$. Assume that algorithm $\mathcal{A}$ satisfies Assumption 3.1 with parameter $\beta_n > 0$, then:*

$$\mathbb{E}\left[\sup_{w \in \mathcal{W}_{S,U}} \left(\mathcal{R}(w) - \widehat{\mathcal{R}}_S(w)\right)\right] \leq 2\mathbb{E}\left[\mathbf{Rad}_{\tilde{S}_J}(\mathcal{W}_{S,U})\right] + 2J\beta_n, \tag{8}$$

*where $\mathbf{Rad}_S(\mathcal{W})$ denotes the Rademacher complexity, whose definition is recalled in Definition A.1.*

The proof is inspired by the data splitting technique used in (Dupuis et al., 2023, Theorem 3.8), which we adapt to our purpose of using the trajectory stability assumption to remove the need for information-theoretic terms.

*Proof.* In the following, we consider tree independent datasets $S, S', S'' \sim \mu_z^{\otimes n}$, with $S = (z_1, \ldots, z_n)$ (and similar notation for $S'$ and $S''$). Up to a slight change in the constant, we can without loss of generality consider $J, K \in \mathbb{N}^\star$ such that $n = JK$. We write the index set as a disjoint union as follows:

$$\{1, \ldots, n\} = \coprod_{k=1}^{K} N_k,$$

with for all $k$, $|N_K| = J$, where $\coprod$ denotes the disjoint union. We also introduce the notation $S_k \in \mathcal{Z}^n$, with:

$$\forall i \in \{1, \ldots, n\}, \ (S_k)_i = \left\{ \begin{array}{ll} z_i', & i \in N_k \\ z_i, & i \notin N_k \end{array} \right.$$

Given two datasets $S_1, S_2 \in \mathcal{Z}^n$, we also consider a measurable mapping $w(\mathcal{W}_{S_1}, S_2)$ (whose existence is satisfied in our setting Molchanov (2017)) such that, almost surely:

$$\omega(\mathcal{W}_{S_1}, S_2) \in \arg\max_{w \in \mathcal{W}_{S_1}} \left( \mathcal{R}(w) - \widehat{\mathcal{R}}_{S_2}(w) \right).$$

Recall that $\mathcal{W}_S \in \mathrm{CL}(\mathbb{R}^d)$ (where $\mathrm{CL}(\mathbb{R}^d)$ denotes the closed subsets of $\mathbb{R}^d$) is a data-dependent random trajectory, which we assume is a random compact set Molchanov (2017). We also denote $\rho_S$ the conditional distribution of $\mathcal{W}_S$ given $S$. Finally we recall the notation:

$$G_S(\mathcal{W}_{S,U}) := \sup_{w \in \mathcal{W}_{S,U}} \left( \mathcal{R}(w) - \widehat{\mathcal{R}}_S(w) \right).$$

We have:

$$\mathbb{E}\left[ G_S(\mathcal{W}_{S,U}) \right] = \mathbb{E}\left[ \mathcal{R}(\omega(\mathcal{W}_{S,U}, S)) - \widehat{\mathcal{R}}_S(\omega(\mathcal{W}_{S,U}, S))) \right]$$

$$= \frac{1}{n} \mathbb{E}\left[ \sum_{k=1}^{K} \sum_{i \in N_k} \left( \ell(\omega(\mathcal{W}_{S,U}, S), z_i'') - \ell(\omega(\mathcal{W}_{S,U}, S), z_i)) \right) \right].$$

By Assumption 3.1 there exists a measurable function $\omega' : \mathrm{CL}(\mathbb{R}^d) \times \mathcal{Z}^n \to \mathbb{R}^d$ such that for any $z \in \mathcal{Z}$

$$\mathbb{E}_U[|\ell(\omega(\mathcal{W}_{S,U}, S), z) - \ell(\omega'(\mathcal{W}_{S_k,U}, \omega(\mathcal{W}_{S,U}), S), z)|] \leq J\beta_n.$$

We now apply the stability assumption on each term of the sum to get:

$$\mathbb{E}\left[ G_S(\mathcal{W}_{S,U}) \right] \leq 2J\beta_n + \frac{1}{n}\mathbb{E}\left[ \sum_{k=1}^{K} \sum_{i \in N_k} \left( \ell(\omega'(\mathcal{W}_{S_k,U}, \omega(\mathcal{W}_{S,U}, S)), z_i'') - \ell(\omega'(\mathcal{W}_{S_k,U}, \omega(\mathcal{W}_{S,U}, S)), z_i)) \right) \right]$$

$$\leq 2J\beta_n + \frac{1}{K}\mathbb{E}\left[ \sum_{k=1}^{K} \sup_{w \in \mathcal{W}_{S_k,U}} \frac{1}{J} \sum_{i \in N_k} \left( \ell(w, z_i'') - \ell(w, z_i)) \right) \right]$$

$$\leq 2J\beta_n + \frac{1}{K}\mathbb{E}\left[ \sum_{k=1}^{K} \sup_{w \in \mathcal{W}_{S,U}} \frac{1}{J} \sum_{i \in N_k} \left( \ell(w, z_i'') - \ell(w, z_i')) \right) \right],$$

where the last equation follows by linearity of the expectation independence by interchanging $z_i$ and $z_i'$ for all $i \in N_k$. By noting that all the terms inside the first sum have the same distribution, we have:

$$\mathbb{E}\left[ G_S(\mathcal{W}_{S,U}) \right] \leq 2J\beta_n + \frac{1}{K} \sum_{k=1}^{K} \mathbb{E}\left[ \sup_{w \in \mathcal{W}_{S,U}} \frac{1}{J} \sum_{i \in N_k} \left( \ell(w, z_i'') - \ell(w, z_i')) \right) \right]$$

$$= 2J\beta_n + \mathbb{E}\left[ \sup_{w \in \mathcal{W}_{S,U}} \frac{1}{J} \sum_{i \in N_1} \left( \ell(w, z_i'') - \ell(w, z_i')) \right) \right],$$

As $\mathcal{W}_{S,U}$ is independent of $S'$ and $S''$, we can apply the classical symmetrization argument. More precisely, let $(\epsilon_1, \ldots, \epsilon_n) \sim \mathcal{U}(\{-1, 1\})^{\otimes n}$ be i.i.d. Rademacher random variables (i.e., centered Bernoulli random variables), independent from the other random variable. We have:

$$\mathbb{E}\left[G_S(\mathcal{W}_{S,U})\right] \leq 2J\beta_n + \mathbb{E}\left[\sup_{w \in \mathcal{W}_{S,U}} \frac{1}{J} \sum_{i \in N_1} \epsilon_i \left(\ell(w, z_i'') - \ell(w, z_i')\right)\right]$$

$$\leq 2J\beta_n + \mathbb{E}\left[\sup_{w \in \mathcal{W}_{S,U}} \frac{1}{J} \sum_{i \in N_1} \epsilon_i \ell(w, z_i'')\right] + \mathbb{E}\left[\sup_{w \in \mathcal{W}_{S,U}} \frac{1}{J} \sum_{i \in N_1} (-\epsilon_i)\ell(w, z_i')\right]$$

$$= 2J\beta_n + 2\mathbb{E}\left[\sup_{w \in \mathcal{W}_{S,U}} \frac{1}{J} \sum_{i \in N_1} \epsilon_i \ell(w, z_i')\right].$$

Therefore, by denoting $\tilde{S}_J = (z_i')_{i \in N_1}$, we obtain by definition of the Rademacher complexity that:

$$\mathbb{E}\left[G_S(\mathcal{W}_{S,U})\right] \leq 2J\beta_n + 2\mathbb{E}\left[\mathbf{Rad}_{\tilde{S}_J}(\mathcal{W}_{S,U})\right],$$

which is the desired result. $\qquad\square$

This result is based on leveraging our stability assumption to enable the use of the classical symmetrization argument Shalev-Shwartz & Ben-David (2014). Compared to previously proposed modified Rademacher complexity bounds, Foster et al. (2019); Sachs et al. (2023), the above theorem has the advantage of directly involving the Rademacher complexity of the data-dependent hypothesis set $\mathcal{W}_S$, which tightens the bound and makes it significantly more empirically relevant.

Compared to previous studies Foster et al. (2019); Dupuis et al. (2024), a drawback of our bound compared to these studies is that it only holds in expectation. Moreover, while we directly involve the data-dependent trajectory $\mathcal{W}_S$, our Rademacher $\mathbf{Rad}_{\tilde{S}}(\mathcal{W}_S)$ term uses an independent copy of the dataset, hence, losing some data dependence compared to the term $\mathbf{Rad}_S(\mathcal{W}_S)$ obtained by Dupuis et al. (2024). That being said, we show that our techniques still recover certain topological complexities, at the cost of requiring an additional Lipschitz continuity assumption. The Rademacher complexity term appearing in Lemma 3.4 uses the data-dependent trajectory $\mathcal{W}_S$ and an independent dataset $\tilde{S}$ of size $J$. Following classical arguments Shalev-Shwartz & Ben-David (2014); Andreeva et al. (2024), we expect this term to behave as $\mathcal{O}(1/\sqrt{J})$, hence, optimizing the value of $J$ yields a bound of order $\mathcal{O}(\beta_n^{1/3})$.

### B.3 COVERING BOUNDS

In this section, we present worst-case generalization bounds over data-dependent random sets based on covering numbers, as they have been defined in Definition A.2. These covering bounds serve as a basis to derive some of our other main results, namely, our generalization bounds based on fractal dimension and $\alpha$-weighted lifetime sums. We believe that these additional contributions are also of independent interest.

The following lemma is a consequence of classical covering argument for Rademacher complexity (Shalev-Shwartz & Ben-David, 2014).

**Lemma B.1** (Covering lemma). *Let $\mathcal{W} \subset \mathbb{R}^d$ be a set with finite diameter. We assume that Assumption 4.2 holds and that the loss $\ell(\cdot, z)$ is $L$-Lipschitz continuous for all $z \in \mathcal{Z}$. Let $m \in \mathbb{N}^*$ and $S \in \mathcal{Z}^m$, we have*

$$\mathbf{Rad}_S(\mathcal{W}) \leq \inf_{\delta > 0} \left(L\delta + B\sqrt{\frac{2\log|N_\delta(\mathcal{W})|}{m}}\right)$$

The argument is very classical, we reproduce it briefly for the sake of completeness.

*Proof.* Let $\delta > 0$ and $N := |N_\delta(\mathcal{W})|$ and let $(x_1, \ldots, x_N)$ be a minimal $\delta$-cover of $\mathcal{W}$. By Lipschitz continuity, we have that:

$$\mathbf{Rad}_S(\mathcal{W}) \leq L\delta + \mathbf{Rad}_S(\{x_1, \ldots, x_N\}).$$

By Massart's lemma (Shalev-Shwartz & Ben-David, 2014, Lemma 26.8), we have that

$$\mathbf{Rad}_S(\mathcal{W}) \leq L\delta + B\sqrt{\frac{2\log N}{m}} = L\delta + B\sqrt{\frac{2\log|N_\delta(\mathcal{W})|}{m}}.$$

We conclude by taking the infimum over $\delta > 0$. $\qquad\square$

To illustrate the power of our framework, we present in the next theorem a data-dependent worst-case bounds for stable random sets in terms of covering numbers of $\mathcal{W}_{S,U}$.

**Theorem B.2.** *Suppose that Assumptions 4.2, 3.1, and 4.1 hold, and that $\mathcal{W}_{S,U}$ is almost surely of finite diameter. Without loss of generality, assume that $\beta_n^{-2/3}$ is an integer divisor of $n^4$. We have:*

$$\mathbb{E}\left[\sup_{w\in\mathcal{W}_{S,U}}\left(\mathcal{R}(w) - \widehat{\mathcal{R}}_S(w)\right)\right] \leq 2\mathbb{E}\left[\inf_{\delta>0}\left(\delta L_{S,U} + \beta_n^{1/3}\left(1 + B\sqrt{2\log|N_\delta(\mathcal{W}_{S,U})|}\right)\right)\right],$$

*where $|N_\delta(\mathcal{W}_{S,U})|$ is covering number, i.e., the minimum number of balls of radius $\delta$ that are necessary to cover $\mathcal{W}_{S,U}$.*

*Proof.* By Lemma 3.4, we have (recall that $G_S$ has been defined in Equation (4)):

$$\mathbb{E}[G_S(\mathcal{W}_{S,U})] \leq 2\mathbb{E}\left[\mathbf{Rad}_{\tilde{S}_J}(\mathcal{W}_{S,U})\right] + 2J\beta_n,$$

with the notations of Lemma 3.4 for $\tilde{S}_J$. By Assumption 4.1, for fixed $(S,U)$ the loss $\ell(\cdot,\mathcal{Z})$ is $L$-Lipschitz continuous for all $z \in \mathcal{Z}$. Therefore, we can apply Lemma B.1 to obtain that

$$\mathbb{E}[G_S(\mathcal{W}_{S,U})] \leq 2\mathbb{E}\left[\inf_{\delta>0}\left(\delta L_{S,U} + B\sqrt{\frac{2\log|N_\delta(\mathcal{W}_{S,U})|}{J}}\right)\right] + 2J\beta_n,$$

We make the choice $J := \beta_n^{-2/3}$, which we assumed for simplicity is an integer dividing $n$, so that it is a valid choice. This immediately gives the result. $\qquad\square$

Covering bounds are widely used for worst-case bounds over data-independent hypothesis sets (i.e., Example 1.4) (Shalev-Shwartz & Ben-David, 2014). Theorem B.2 extends these results to data-dependent hypothesis sets. Theorem B.2 also improves over the data-dependent covering bounds of Dupuis et al. (2024, Corollary 33) by avoiding the use of complicated information-theoretic terms and by allowing the Lipschitz constant to be defined only on the data-dependent set $\mathcal{W}_{S,U}$.

### B.4 PROOF OF THEOREM 4.3

We now present the proof of Theorem 4.3. This theorem is based on our key technical lemma (Lemma 3.4), the covering lemma (Lemma B.1), and a limit argument that is inspired from previous works (Simsekli et al., 2020; Dupuis & Viallard, 2023).

**Theorem 4.3.** *Suppose that Assumptions 4.2, 3.1, and 4.1 hold, and that $\mathcal{W}_{S,U}$ is a.s. of finite diameter. Without loss of generality, assume that $\beta_n^{-2/3}$ is an integer divisor of $n$. There exists $\delta_n > 0$ such that for all $\delta < \delta_n$*

$$\mathbb{E}\left[\sup_{w\in\mathcal{W}_{S,U}}\left(\mathcal{R}(w) - \widehat{\mathcal{R}}_S(w)\right)\right] \leq 2\mathbb{E}\left[\frac{B}{n} + \delta L_{S,U} + \beta_n^{1/3}\left(1 + B\sqrt{4\overline{\dim}_\mathrm{B}(\mathcal{W}_{S,U})\log\frac{1}{\delta}}\right)\right],$$

*where $\overline{\dim}_\mathrm{B}(\mathcal{W}_{S,U})$ is the upper box-counting dimension (Falconer, 2013), which we define in the appendix in Definition A.4. We refer to the Appendix B.4 for a discussion of the parameter $\delta_n$.*

*Proof.* Let $(\Omega, \mathcal{F}_U, \mathbb{P}_U)$ denote the probability space to which belongs $U$, with $\mathbb{P}_U$ denoting the law of $U$. We also denote $\mathcal{F}$ the $\sigma$-algebra associated with the data space $\mathcal{Z}$. As $\mathcal{W}_{S,U}$ is almost-surely

---

[4] Not making this assumption would only lead to minor changes in the bound

of finite diameter, by definition of the upper box-counting dimension, we have $\mu_z^{\otimes n} \otimes \mathbb{P}_U$-almost surely that:

$$\overline{\dim}_{\mathrm{B}}(\mathcal{W}_{S,U}) := \limsup_{\delta \to 0} \frac{\log |N_\delta(\mathcal{W}_{S,U})|}{\log(1/\delta)} = \lim_{\delta \to 0} \sup_{0 < r < \delta} \frac{\log |N_r(\mathcal{W}_{S,U})|}{\log(1/r)} =: \lim_{\delta \to 0} f_\delta(\mathcal{W}_{S,U}).$$

Let $(\delta_k)_{k \geq 0}$ be a decreasing positive sequence in $(0,1)$ with $\delta_k \to 0$. Let also $\gamma > 0$. By Egoroff's theorem (Bogachev, 2007), there exists a set $\Omega_\gamma \in \mathcal{F}^{\otimes n} \otimes \mathcal{F}_U$ such that $\mu_z^{\otimes n} \otimes \mathbb{P}_U(\Omega_\gamma) \geq 1 - \gamma$, and such that the convergence of $\log f_{\delta_k}(\mathcal{W}_{S,U}) \to_k \log \overline{\dim}_{\mathrm{B}}(\mathcal{W}_{S,U})$ is uniform on $\Omega_\gamma$. Therefore, there exists $k_{n,\gamma} > 0$ such that for all $k \geq k_{n,\gamma}$, we have for $(S,U) \in \Omega_\gamma$

$$\log f_{\delta_k}(\mathcal{W}_{S,U}) \leq \log \overline{\dim}_{\mathrm{B}}(\mathcal{W}_{S,U}) + \log 2.$$

Therefore, for all $0 < \delta < \delta_k$, we have by taking the exponential that for $(S,U) \in \Omega_\gamma$:

$$\log |N_\delta(\mathcal{W}_{S,U})| \leq 2 \log(1/\delta) \overline{\dim}_{\mathrm{B}}(\mathcal{W}_{S,U}).$$

Now we use Lemma 3.4 to get that for all $J$ we have

$$\mathbb{E}\left[G_S(\mathcal{W}_{S,U})\right] \leq 2\beta_n J + 2\mathbb{E}_{S,U,\tilde{S}}\left[\mathbf{Rad}_{\tilde{S}_J}(\mathcal{W}_{S,U})\right]$$

$$\leq 2\beta_n J + 2B\mathbb{E}_{S,U,\tilde{S}}\left[\mathbb{1}_{\overline{\Omega_\gamma}}(S,U)\right] + 2\mathbb{E}_{S,U,\tilde{S}}\left[\mathbb{1}_{\Omega_\gamma}(S,U)\mathbf{Rad}_{\tilde{S}_J}(\mathcal{W}_{S,U})\right]$$

$$\leq 2\beta_n J + 2B\gamma + 2\mathbb{E}_{S,U,\tilde{S}}\left[\mathbb{1}_{\Omega_\gamma}(S,U)\mathbf{Rad}_{\tilde{S}_J}(\mathcal{W}_{S,U})\right].$$

We make the choice $J := \beta_n^{-2/3}$, which we assumed for simplicity is an integer dividing $n$, so that it is a valid choice. We conclude the proof by Lemma B.1 and setting $\gamma := 1/n$ and $\delta_n := k_{n,1/n}$. $\square$

**Remark B.1.** The careful reader might notice that the proof above requires some measure-theoretic conditions to hold, in particular, the measurability of the mappings $(S,U) \mapsto \mathbf{Rad}_S(\mathcal{W}_{S,U})$ and $(S,U) \mapsto |N_\delta(\mathcal{W}_{S,U})|$. These aspects have been extensively studied in existing works on covering / fractal bounds on random sets (Dupuis et al., 2023; 2024). In our paper, we implicitly assume these conditions to be satisfied for the sake of simplicity, and refer to these works for more details. This choice is justified as our main application regards finite trajectories (Example 1.1), where all this conditions are satisfied.

**Remark B.2.** The parameter $\delta_n$ appearing in the statement of Theorem 4.3 might seem intriguing. It can be seen from the proof that this parameters quantifies the uniformity in $n$ of the limit defining the upper box-counting dimension (i.e., if this convergence is uniform in $n$, then $\delta$ is independent of $n$). Such a parameter already appears in (Dupuis et al., 2023) in a similar context and it was shown in (Dupuis et al., 2024) that $\delta$ can be made independent of $n$ under a convergence assumption of the random sets distributions. We refer to these works for further details.

## B.5 Proof of Theorem 4.4

Before presenting the proof of Theorem 4.4, we present two technical lemma, whose proof can be found in Andreeva et al. (2024). The first lemma relates the Rademacher complexity to the $\alpha$-weighted lifetime sums.

**Lemma B.3.** *Let $m \in \mathbb{N}^\star$, $S \in \mathcal{Z}^m$, and $\mathcal{W} \subset \mathbb{R}^d$ be a finite set. Assume that Assumption 4.2 and that $\ell(\cdot, z)$ is $L$-Lipschitz continuous on $\mathcal{W}$ for all $z \in \mathcal{W}$. For any $\alpha \in (0,1]$, we have:*

$$\mathbf{Rad}_S(\mathcal{W}) \leq \frac{B}{\sqrt{m}} + B\sqrt{\frac{2\log(1 + K_{n,\alpha}\mathbf{E}^\alpha(\mathcal{W}))}{m}},$$

*with $K_{n,\alpha} := 2(2L\sqrt{n}/B)^\alpha$.*

*Proof.* This result is a particular case of a bound by Andreeva et al. (2024) as part of the proof of their Theorem 3.4. The assumptions of (Andreeva et al., 2024, Theorem 3.4) are satisfied by Assumption 4.2, the Lipschitz continuity assumption, and by considering the Euclidean distance of $\mathbb{R}^d$ as a pseudometric on $\mathcal{W}$. $\square$

The second lemma relates the Rademacher complexity to the positive magnitude introduced by Andreeva et al. (2024).

**Lemma B.4.** *Let $m \in \mathbb{N}^\star$, $S \in \mathcal{Z}^m$, and $\mathcal{W} \subset \mathbb{R}^d$ be a finite set. Assume that Assumption 4.2 holds and that $\ell(\cdot, z)$ is $L$-Lipschitz continuous on $\mathcal{W}$ for all $z \in \mathcal{W}$. Then, we have for any $\lambda > 0$ that:*

$$\mathbf{Rad}_S(\mathcal{W}) \leq \frac{\lambda B^2}{2m} + \frac{1}{\lambda} \log\left(\mathbf{PMag}(L\lambda\mathcal{W})\right).$$

This result was obtained in the proof of (Andreeva et al., 2024, Theorem 3.5) under a quite general Lipschitz regularity condition. For the sake of completeness, we present the proof in our particular case, as it might be more concrete for some readers.

*Proof.* Let us fix $S := (z_1, \ldots, z_m) \in \mathcal{Z}^m$.

It was shown by Meckes (2015) that a finite set $\mathcal{W}$ has magnitude. More precisely, for any $\lambda > 0$ there exists a vector $\beta_s : \mathcal{W} \to \mathbb{R}$, such that for any $a \in \mathcal{W}$ we have:

$$1 = \sum_{b \in \mathcal{W}} e^{-\lambda\|a-b\|} \beta_\lambda(b).$$

Let $\epsilon = (\epsilon_1, \ldots, \epsilon_m) \sim \mathcal{U}(\{-1, 1\})^{\otimes m}$ be Rademacher random variables (see Definition A.1). By Lipschitz continuity, we have for all $a \in \mathcal{W}$ that

$$1 \leq \sum_{b \in \mathcal{W}} e^{-\lambda L\|a-b\|} \beta_{\lambda L}(b)_+$$

$$\leq \sum_{b \in \mathcal{W}} \exp\left(-\frac{\lambda}{m} \sum_{i=1}^m |\ell(a, z_i) - \ell(b, z_i)|\right) \beta_{\lambda L}(b)_+$$

$$\leq \sum_{b \in \mathcal{W}} \exp\left(-\frac{\lambda}{m} \sum_{i=1}^m \epsilon_i(\ell(a, z_i) - \ell(b, z_i))\right) \beta_{\lambda L}(b)_+.$$

Therefore:

$$\exp\left(\frac{\lambda}{m} \sum_{i=1}^m \epsilon_i \ell(a, z_i)\right) \leq \sum_{b \in \mathcal{W}} \exp\left(\frac{\lambda}{m} \sum_{i=1}^m \epsilon_i \ell(b, z_i)\right) \beta_{\lambda L}(b)_+. \tag{10}$$

We apply it to the following choice of $a$:

$$a = a(\epsilon) := \arg\max_{a \in \mathcal{W}} \frac{\lambda}{m} \sum_{i=1}^m \epsilon_i \ell(a, z_i).$$

Thus, by Jensen's inequality, we have:

$$\mathbf{Rad}_S(\mathcal{W}) = \mathbb{E}_\epsilon\left[\max_{a \in \mathcal{W}} \frac{1}{m} \sum_{i=1}^m \epsilon_i \ell(a, z_i)\right]$$

$$= \mathbb{E}_\epsilon\left[\frac{1}{m} \sum_{i=1}^m \epsilon_i \ell(a(\epsilon), z_i)\right]$$

$$\leq \frac{1}{\lambda} \log \mathbb{E}_\epsilon\left[\exp\left(\frac{\lambda}{m} \sum_{i=1}^m \epsilon_i \ell(a(\epsilon), z_i)\right)\right]$$

$$\leq \frac{1}{\lambda} \log \mathbb{E}_\epsilon\left[\sum_{b \in \mathcal{W}} \exp\left(\frac{\lambda}{m} \sum_{i=1}^m \epsilon_i \ell(b, z_i)\right) \beta_{\lambda L}(b)_+\right].$$

By Assumption 4.2, independence, and Hoeffding's lemma, we obtain:

$$\mathbf{Rad}_S(\mathcal{W}) \leq \frac{1}{\lambda} \log \sum_{b \in \mathcal{W}} \prod_{i=1}^m \exp\left(\frac{4B^2\lambda^2}{8m^2}\right) \beta_{\lambda L}(b)_+$$

$$= \frac{1}{\lambda} \log\left(\exp\left(\frac{B^2\lambda^2}{2m}\right) \sum_{b \in \mathcal{W}} \beta_{\lambda L}(b)_+\right)$$

$$= \frac{\lambda B^2}{2m} + \frac{1}{\lambda} \log \mathbf{PMag}(L\lambda \cdot \mathcal{W}),$$

which is the desired result. $\square$

We can finally prove Theorem 4.4.

**Theorem 4.4.** *Suppose that Assumptions 4.2, 3.1, and 4.1 hold. We further assume that the set $\mathcal{W}_{S,U}$ is almost surely finite. Without loss of generality, assume that $\beta_n^{-2/3}$ is an integer divisor of $n$. Then, for any $\alpha \in (0,1]$, we have*

$$\mathbb{E}\left[\sup_{w \in \mathcal{W}_{S,U}} (\mathcal{R}(w) - \widehat{\mathcal{R}}_S(w))\right] \leq \beta_n^{1/3}\left(2 + 2B + 2B\mathbb{E}\left[\sqrt{2\log(1 + K_{n,\alpha}\mathbf{E}^\alpha(\mathcal{W}_{S,U}))}\right]\right),$$

*where $K_{n,\alpha} := 2(2L_{S,U}\sqrt{n}/B)^\alpha$. Moreover, for any $\lambda > 0$, let $s(\lambda) := \lambda L_{S,U}\beta_n^{-1/3}/B$. We have*

$$\mathbb{E}\left[\sup_{w \in \mathcal{W}_{S,U}} (\mathcal{R}(w) - \widehat{\mathcal{R}}_S(w))\right] \leq \beta_n^{1/3}\left(2 + \lambda B + \frac{2B}{\lambda}\mathbb{E}\left[\log \mathbf{PMag}\left(s(\lambda) \cdot \mathcal{W}_{S,U}\right)\right]\right).$$

*Proof.* **Persistent homology bound**. By Lemma 3.4, we have (recall that $G_S$ has been defined in Equation (4)):

$$\mathbb{E}\left[G_S(\mathcal{W}_{S,U})\right] \leq 2\mathbb{E}\left[\mathbf{Rad}_{\tilde{S}_J}(\mathcal{W}_{S,U})\right] + 2J\beta_n,$$

with the notations of Lemma 3.4 for $\tilde{S}_J$. Now we can apply Lemma B.3 to obtain that:

$$\mathbb{E}\left[G_S(\mathcal{W}_{S,U})\right] \leq 2\left(\frac{B}{\sqrt{J}} + B\mathbb{E}\left[\sqrt{\frac{2\log(1 + K_{n,\alpha}\mathbf{E}^\alpha(\mathcal{W}_{S,U}))}{J}}\right]\right) + 2J\beta_n,$$

We make the choice $J := \beta_n^{-2/3}$, which we assumed for simplicity is an integer dividing $n$, so that it is a valid choice. This immediately gives the result.

**Positive magnitude bound** By Lemma 3.4, we have (recall that $G_S$ has been defined in Equation (4)):

$$\mathbb{E}\left[G_S(\mathcal{W}_{S,U})\right] \leq 2\mathbb{E}\left[\mathbf{Rad}_{\tilde{S}_J}(\lambda L_{S,U}\mathcal{W}_{S,U})\right] + 2J\beta_n,$$

with the notations of Lemma 3.4 for $\tilde{S}_J$. Now we can apply Lemma B.3 to obtain that:

$$\mathbb{E}\left[G_S(\mathcal{W}_{S,U})\right] \leq 2\mathbb{E}\left[\inf_{\lambda > 0}\left(\frac{\lambda B^2}{2J} + \frac{1}{\lambda}\log\left(\mathbf{PMag}(\lambda L_{S,U}\mathcal{W}_{S,U})\right)\right) + 2J\beta_n\right].$$

We make the change of variable $\lambda \to \lambda\sqrt{J}/B$, which gives that for any $\lambda > 0$, we have:

$$\mathbb{E}\left[G_S(\mathcal{W}_{S,U})\right] \leq \mathbb{E}\left[\frac{\lambda B}{\sqrt{J}} + \frac{2B}{\lambda\sqrt{J}}\log \mathbf{PMag}(\frac{\lambda L_{S,U}\sqrt{J}}{B}\mathcal{W}_{S,U}) + 2J\beta_n\right].$$

We make the choice $J := \beta_n^{-2/3}$, which we assumed for simplicity is an integer dividing $n$, so that it is a valid choice. This immediately gives the result. $\square$

To complete this section, we also show Lemma 3.2

### B.6 Proof of Lemma 3.2

**Lemma 3.2.** *Consider a fixed $K \in \mathbb{N}^\star$, $1 \leq k \leq K$, and algorithms[5] $\mathcal{A}_k(S,U) := w_k$. Let $\mathcal{A}(S,U) = \mathcal{W}_{S,U} = \{\mathcal{A}_k(S,U)\}_{k=1}^K$ be an algorithm in the sense of Equation (3). Assume that for all $k$, $\mathcal{A}_k$ is $\delta_k$-uniformly argument-stable in the sense of Definition 2.1 and that the loss $\ell(w,z)$ is $L$-Lipschitz-continuous in $w$. Then, $\mathcal{A}$ is random set stable with parameter at most $\beta_n = L\sum_{k=1}^K \delta_k$.*

*Proof.* Consider a data-dependent selection of $\mathcal{A}$, denoted $\omega(\mathcal{W}, S)$, in the sense of Definition 3.1. We consider a measurable mapping $\omega' : \mathrm{CL}(\mathbb{R}^d) \times \mathbb{R}^d \to \mathbb{R}^d$ such that for any closed set $\mathcal{W} \subset \mathbb{R}^d$ and any $w \in \mathbb{R}^d$, we have:

$$\omega'(\mathcal{W}, w) \in \arg\min_{w' \in \mathcal{W}}\left(\sup_{z \in \mathcal{Z}} |\ell(w, z) - \ell(w', z)|\right).$$

---

[5] $\mathcal{A}_k$ is an algorithm with values in $\mathbb{R}^d$, i.e., its output is the $k$-th iterate of the optimization algorithm.

Such a mapping can be constructed in our setting where the sets are compact and the loss continuous. Consider two datasets $S, S'$ in $\mathcal{Z}^n$ differing by $J$ elements. Let us denote:

$$\varepsilon := \sup_{z \in \mathcal{Z}} \mathbb{E}_U \left[ |\ell(\omega(\mathcal{W}_{S,U}, S), z) - \ell(\omega'(\mathcal{W}_{S',U}, \omega(\mathcal{W}_{S,U}, S)), z)| \right]$$

We use the definition of $\omega'$ and the Lipschitz-continuity of $\ell$ to obtain that:

$$\varepsilon \leq \sup_{z \in \mathcal{Z}} \mathbb{E}_U \left[ \max_{1 \leq k \leq K} |\ell(\mathcal{A}_k(S, U), z) - \ell(\omega'(\mathcal{W}_{S',U}, \mathcal{A}_k(S, U)), z)| \right]$$

$$\leq \sup_{z \in \mathcal{Z}} \sum_{k=1}^{K} \mathbb{E}_U \left[ |\ell(\mathcal{A}_k(S, U), z) - \ell(\omega'(\mathcal{W}_{S',U}, \mathcal{A}_k(S, U)), z)| \right]$$

$$\leq \sum_{k=1}^{K} \mathbb{E}_U \left[ \sup_{z \in \mathcal{Z}} |\ell(\mathcal{A}_k(S, U), z) - \ell(\omega'(\mathcal{W}_{S',U}, \mathcal{A}_k(S, U)), z)| \right]$$

$$\leq \sum_{k=1}^{K} \mathbb{E}_U \left[ \sup_{z \in \mathcal{Z}} |\ell(\mathcal{A}_k(S, U), z) - \ell(\mathcal{A}_k(S', U), z)| \right]$$

$$\leq L \sum_{k=1}^{K} \mathbb{E}_U \left[ \|\mathcal{A}_k(S, U) - \mathcal{A}_k(S', U)\| \right]$$

$$= LJ \sum_{k=1}^{K} \delta_k,$$

where the last step follows by applying the stability assumption $J$ times and the triangle inequality. The result follows by definition of the random trajectory stability assumption. $\square$

## C  ADDITIONAL EXPERIMENTAL DETAILS

This section expands on our experimental analyis, providing additional discussion of the methodology, implementation details, and technical setup.

### C.1  IMPLEMENTATION DETAILS

We implement training and validation loops in PyTorch Lightning Version 2.4 using pytorch version 2.4.1. Both models are trained using the AdamW optimizer with a cosine annealing learning rate scheduler. The training was stopped after reaching an accuracy of $95\%$. No regularization techniques were used to prevent confounding factors in each fine-tuning task.

**Vision Transformer (ViT)**. The ViT is configured for image classification on the CIFAR100 dataset. For the Vision Transformer on CIFAR100, we utilize a model with approximately 1.2 million parameters. This represents a reduction in size compared to the ViT experiments in Andreeva et al. (2024), as the empirical analyses we conduct require substantially more experiments due to the expansion of training set size $n$ and thus storage of weight iterates.

The ViT processes 32x32 pixel images with 3 color channels, and a patch size of 4x4. The transformer backbone consists of 6 layers, 8 attention heads and an MLP dimension of 512. Dropout rates are set to 0.1 within the transformer blocks. Training is performed with a base learning rate of 0.0005, decaying to a minimum of $10^{-5}$ via cosine annealing.

**GraphSage**. We analyze GraphSAGE Hamilton et al. (2017) on the graph classification task for MNISTSuperpixels Monti et al. (2017). This model consists of approximately 0.2 million parameters. It processes graphs where each node has a single input feature. These features are initially projected to a hidden dimension of 128. For graph-level predictions, node embeddings are aggregated using 'mean' global pooling, followed by a linear layer for classification into 10 classes. The model uses 2-dimensional positional encodings (representing x,y coordinates), which are added to the node features. We use 4 GraphSAGE convolutional layers with a dropout rate of 0.05, a base learning rate of 0.001, and weight decay of 0.0005 to achieve an adequate generalization gap.

## C.2 STABILITY ANALYSIS

We first elaborate on our procedure to estimate random set stability in Section 5. The overall algorithm, a proxy to asses Section 5, is summarized in Algorithm 1. Stability experiments were performed with a fixed learning rate ($\eta$) and batch size ($b$), while varying random seeds (e.g. $\{100, 200, 300, 400, 500\}$. The same setup is then conducted twice: once with the replacement set of size $J = 50$, and once without. Since Assumption 3.1 requires taking the supremum over all samples $Z \in \mathcal{Z}$, we approximate it by evaluating the stability parameter on 500 samples seen during training and 500 unseen samples. In Table 1, we report the maximum value across both sets.

---

**Algorithm 1** Stability Parameter Estimation

---

**Require:** $S = \{z_1, \ldots, z_n\}$
**Require:** $S_J = \{z'_1, \ldots, z'_J\}$
**Require:** $S_J \cap S = \emptyset$
**Require:** $J \leq n$
  1: **procedure** PROCEDURE(Training)
  2:      Let $I = \{i_1, \ldots, i_J\}$ be an uniformly random subset of $\{1, \ldots, n\}$      ▷ Indices to replace
  3:      $S_{new} := S \setminus \{z_i : i \in I\} \cup S_J := \{z_{new,1}, \ldots, z_{new,n}\}$
  4:      Generate $\mathcal{W}_{S,U}$ with $|\mathcal{W}_{S,U}| = T$
  5:      Generate $\mathcal{W}_{S_{new},U}$ with $|\mathcal{W}_{S_{new},U}| = T$

  6: **procedure** PROCEDURE(Estimation)
  7:      $M = \min\{n, 500\}$
  8:      $L = \{j_1, \ldots, j_M\}$ be an uniform random subset of $\{1, \ldots, n\}$
  9:      $S_{eval} := \{z_{new,i} : i \in L\} \setminus S_J$
10:      $N := |S_{eval}| = M - J$
11:
12:      Initialize $A^1, \ldots, A^N \in \mathbb{R}^{T \times T}$
13:      Let $w_l \in \mathcal{W}_{S,U} \; \forall l \in \{1, \ldots, T\}$
14:      Let $w_{new,l} \in \mathcal{W}_{S_{new},U} \; \forall l \in \{1, \ldots, T\}$
15:      Let $\ell(\cdot, \cdot)$ denote the used loss function.
16:
17:      For all $i \in \{1, \ldots, N\}$ compute
18:      **for** $j \leftarrow 1$ to $T$ **do**
19:        **for** $l \leftarrow 1$ to $T$ **do**
20:          $A^i_{j,l} := |\ell(w_j, Z_i) - \ell(w_{new,l}, Z_i)|$      ▷ $Z_i \in S_{eval}$
21:      Now we compute $A := \frac{1}{N} \sum_{i=1}^{N} A^i$
22:
23:      Now we take the minimum over each row in matrix $A$
24:      Initialize $a \in \mathbb{R}^T$ and compute
25:      **for** $m \leftarrow 1$ to $T$ **do** $a_m = \min_{1 \leq l \leq T} A_{m,l}$
26:      The algorithm ends with $\max_{1 \leq h \leq m} a_h$

---

## C.3 BOUND ESTIMATION

In the first part of Section 5, we estimate the Rademacher complexity bound stated in Lemma 3.4. In particular, we optimize the quantity

$$2\sqrt{\frac{2 \log T}{J}} \; + \; 2J\beta_n,$$

which is minimized at

$$J^* \; = \; 2^{-\frac{1}{3}} \, \beta_n^{-\frac{2}{3}} \, (\log T)^{-\frac{1}{3}}.$$

Since $J$ must be a divisor of the sample size $n$ (see Lemma 3.4), we define the effective choice of $J$ as

$$J \; = \; \underset{d|n}{\arg\min} \, |d - J^*|.$$

## C.4 Topological Complexity Analysis

In this section we detail our approach for weight trajectory analysis in terms of topological quantities, particularly experiments which demonstrate the relationship between stability and our topological complexity.

Following Andreeva et al. (2024), we fine-tune each model from a local minimum for an additional 5000 iterations after training to convergence. We use a reduced parameter grid of $4 \times 4$ for $\eta$ and $b$ (Andreeva et al. (2024) use $6 \times 6$), due to computational constraints arising from the varying the training set size $n$. Detailed hyperparameters are listed in Table 2.

| Hyperparameter | Vision Transformer | GraphSage |
|---|---|---|
| Sample Size ($n$) | $\{100, 500, 1000, 5000, 10000\}$ | $\{100, 500, 1000, 5000, 10000\}$ |
| Learning Rate ($\eta$) | $\{10^{-6}, 10^{-5}, 5 \times 10^{-5}, 10^{-4}\}$ | $\{10^{-5}, 10^{-4}, 5 \times 10^{-4}, 10^{-3}\}$ |
| Batch Size ($b$) | $\{32, 64, 128, 256\}$ | $\{32, 64, 128, 256\}$ |

Table 2: Summary of hyperparameter configurations for Vision Transformer and GraphSage models.

## C.5 Hardware and Runtime

Models are trained on a cluster with 40GB A40/A100 GPUs; although the experiments are reproducible with lower memory requirements. For our grid experiments, computing topological features alongside hyperparameter variations (sample size, learning rate, and batch size) demanded approximately 57 GPU hours for GraphSage and 140 GPU hours for the Vision Transformer (ViT). In contrast, the stability analysis required about 50 total GPU hours.

# D Additional Empirical Results

In this section, we present additional empirical results. The first part complements the second part of Section 5. In the main text, we focused on the $\alpha$-weighted lifetime sum in Theorem 4.4. The second bound, given in terms of the positive magnitude $\mathbf{PMag}$ will be discussed here.

Motivated by the dependence of the stability parameter in Assumption 3.1 on the sample size $n$, we also investigate the topological complexity measures introduced in Theorem 4.4. To the best of our knowledge, this behavior has not been addressed in previous work. We study it here to provide a more complete picture of the $\alpha$-weighted lifetime sum and the positive magnitude.

## D.1 Interplay between Stability and $\mathbf{C}(\mathcal{W}_{S,U})$

In this subsection, in addition to the results presented in Section 5, we provide the results specifically for the *positive magnitude*. The second bound in Theorem 4.4 depends on the scaling parameter $s(\lambda)$. Recall that in Theorem 4.4, the parameter $s(\lambda)$ can be chosen arbitrarily. The theoretically justified choice is $s(\lambda) = \mathcal{O}(n^{-1/3})$ in order to obtain the optimal convergence rate. Nevertheless, alternative choices of $s(\lambda)$ have also led to remarkable correlation results in previous work Andreeva et al. (2024).We focus on the setting $s(\lambda) = n^{-1/3}$ with $n \in \{100, 10^4\}$.

**Results and Discussion**. We observe the same behavior as in the main text (Section 5). We note that the effects for GraphSage Figure 4 and Figure 5 are less pronounced than for the ViT (Figure 6 and Figure 7); nonetheless, the general trend remains visible. Interestingly, for $s(\lambda) = 0.01$, we observe a different behavior. While for higher values of $s(\lambda)$ the steepness varies with the sample size, as predicted by Theorem 4.4 and also observed for $\mathbf{E}^1$, the steepness remains consistent across different values of $n$.

The same behavior for $\mathbf{PMag}(s(\lambda) \cdot \mathcal{W}_{S,U})$, with $s(\lambda) = n^{\frac{1}{3}}$, can be explained theoretically. In Theorem 4.4, we observe the product structure. Indeed, Theorem 4.4 asserts that $\log \mathbf{E}^1(\mathcal{W}_{S,U})$ should be approximately of order at least $\beta_n^{-1/3} G_S(\mathcal{W}_{S,U})$, which corresponds to $n^{1/3} G_S(\mathcal{W}_{S,U})$ when $\beta_n = \Theta(1/n)$. Therefore, our experimental results provide strong empirical support for Theorem 4.4 both for $\mathbf{E}^1$ and $\mathbf{PMag}$.

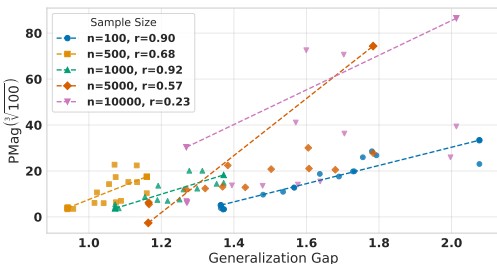

Figure 4: Variation of $\mathbf{PMag}(\sqrt[3]{100} \cdot \mathcal{W}_{S,U})$ with sample size $n$ for the model **GraphSage**. Pearson correlation coefficients $r$ are reported for each subgroup.

Figure 5: Variation of $\mathbf{PMag}(\sqrt[3]{10000} \cdot \mathcal{W}_{S,U})$ with sample size $n$ for the model **GraphSage**. Pearson correlation coefficients $r$ are reported for each subgroup.

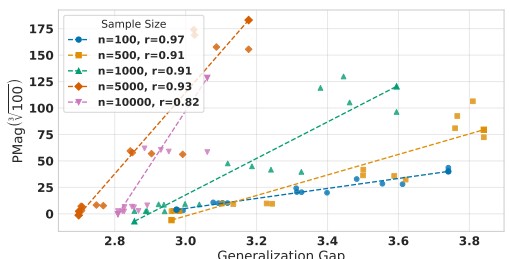
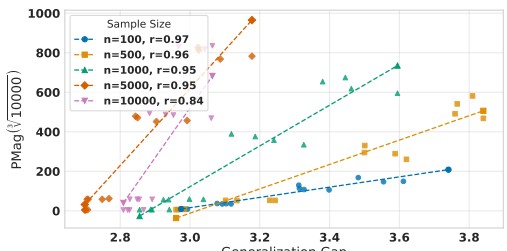

Figure 6: Variation of $\mathbf{PMag}(\sqrt[3]{100})$ with sample size $n$ for the model **ViT**. Pearson correlation coefficients $r$ are reported for each subgroup.

Figure 7: Variation of $\mathbf{PMag}(\sqrt[3]{10000})$ with sample size $n$ for the model **ViT**. Pearson correlation coefficients $r$ are reported for each subgroup.

## D.2 Complexity Measures $\mathbf{C}(\mathcal{W}_{S,U})$ as functions of the sample size $n$

While the main write-up in Section 5 focused on order of the bound values of Lemma 3.4 and overall behavior, in this section we investigate the behavior of the topological complexity measure introduced in Andreeva et al. (2024). In particular, in Theorem 4.4 we recover the same complexity measures $\mathbf{C}(\mathcal{W}_{S,U})$ within our stability framework. Motivated by the dependence of the stability parameter in Assumption 3.1 on the sample size $n$, we investigate the *positive magnitude* and the *$\alpha$-weigthed lifetime sum* with respeect to the training data size.

**Experimental Design**. Similarly to Section 5, we investigate the behavior of two conceptually different models: a Vision Transformer (**ViT**) and **GraphSage**. The implementation details remain consistent with the main experimental setting. In particular, we analyze the behavior of $\mathbf{E}^1(\mathcal{W}_{S,U})$ as a function of the sample size $n$, for fixed learning rates $\eta = 10^{-4}$ and $\eta = 10^{-5}$ for ViT, and $\eta = 10^{-3}$ and $\eta = 10^{-4}$ for GraphSage, across all batches. Our focus is on $\mathbf{E}^1(\mathcal{W}_{S,U})$ and $\mathbf{PMag}(s(\lambda) \cdot \mathcal{W}_{S,U})$, with $s(\lambda) = n^{\frac{1}{3}}$ and $n \in \{100, 10^4\}$.

**Results and Discussion**. For fixed learning rates, we observe that both $\mathbf{E}^1$ and $\mathbf{PMag}$ increase with increasing $n$ and appear to stabilize for large values of $n$ (see Figures 9, 8, 11, 10, 13, 12 . This suggests a relationship between the sample size and the topology of the training trajectory, a phenomenon that has not been previously highlighted in the literature.

At first glance, the increase of $\mathbf{E}^1(W_S, U)$ with $n$ may seem counterintuitive, since we typically expect generalization error (and related bounds) to decrease as the sample size grows. However, this behavior is consistent with theory: the $\alpha$-weigthed lifetime sum is expected to grow at most polynomially with $n$. The positive magnitude remains bounded in terms of the number of iterations.

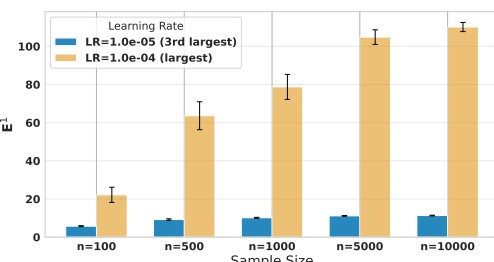

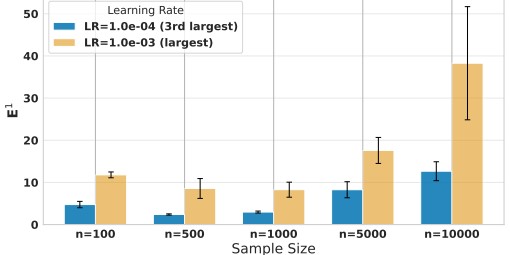

Figure 8: Evolution of $\mathbf{E}^1(\mathcal{W}_{S,U})$ with increasing sample size n, for learning rates $\eta = 10^{-5}$ and $\eta = 10^{-4}$ (**ViT**).

Figure 9: Evolution of $\mathbf{E}^1(\mathcal{W}_{S,U})$ with increasing sample size n, for learning rates $\eta = 10^{-3}$ and $\eta = 10^{-4}$ (**GraphSage**).

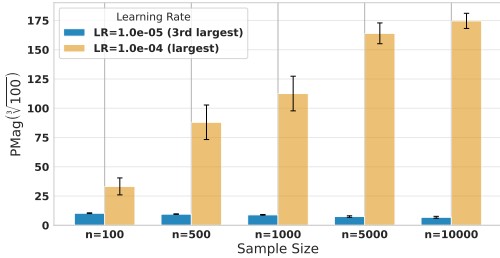

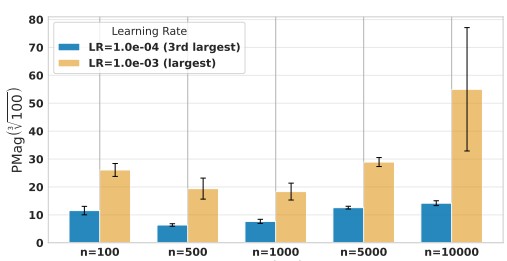

Figure 10: Evolution of $\mathbf{PMag}(\sqrt[3]{100} \cdot \mathcal{W}_{S,U})$ with increasing sample size n, for learning rates $\eta = 10^{-5}$ and $\eta = 10^{-4}$ (**ViT**).

Figure 11: Evolution of $\mathbf{PMag}(\sqrt[3]{100} \cdot \mathcal{W}_{S,U})$ with increasing sample size n, for learning rates $\eta = 10^{-3}$ and $\eta = 10^{-4}$ (**GraphSage**).

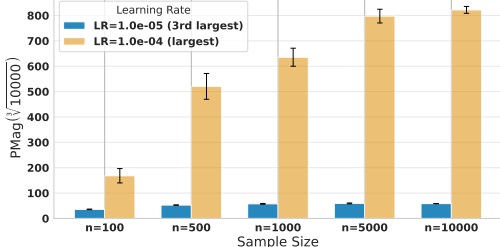

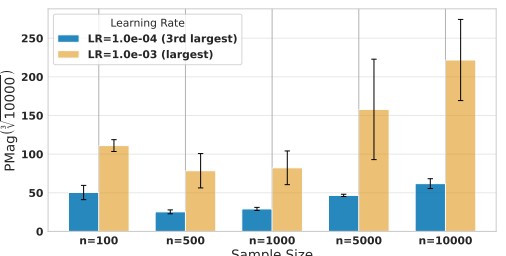

Figure 12: Evolution of $\mathbf{PMag}(\sqrt[3]{10000} \cdot \mathcal{W}_{S,U})$ with increasing sample size n, for learning rates $\eta = 10^{-4}$ and $\eta = 10^{-4}$ (**ViT**).

Figure 13: Evolution of $\mathbf{PMag}(\sqrt[3]{10000} \cdot \mathcal{W}_{S,U})$ with increasing sample size n, for learning rates $\eta = 10^{-3}$ and $\eta = 10^{-4}$ (**GraphSage**).

