# OpenReview forum: "Stability, Complexity and Data-Dependent Worst-Case Generalization Bounds"
_ICLR.cc/2026/Conference — Submitted to ICLR 2026_

### Official Review · Reviewer_PcQB · 2025-10-30

**Soundness:** 3
**Presentation:** 3
**Contribution:** 3
**Rating:** 6
**Confidence:** 3

**Summary:**

This paper focuses on worst-case generalization on the weight trajectory/data-dependent set. The paper identifies that existing worst-case bounds have the following problems: They involve intractable information-theoretic or combinatorial nature quantities that are hard to compute, or they ignore the randomness of the algorithms.
This paper then develops random-set stability that considers algorithmic randomness and proves worst-case generalization gap can be bounded together by the stability and Rademacher complexity or fractal-based complexities. Empirical results show that the bounds are non-vacuous in many cases and successfully predict the interplay between the stability and fractal-based complexities.

**Strengths:**

- This paper reveals valuable problems in existing works of worst-case generalization, and thereby is well motivated. This problem is interested not only for worst-case bounds, but also in other topics, eg, generalization bounds using mutual information.
- This paper provides novel stability-based worst-case bounds with tractable quantities. The bounds not only recover several classic bounds, but also can be used with recent fractal-based complexities.
- In experiments, the bounds are non-vacuous in many cases.
- The paper is well written and clear to follow.

**Weaknesses:**

1. The auto-satisfaction of local Lipschitzness in Line 315 seems incorrect because the Lipschitzness is not local wrt $z$. For example, for loss $\\ell(w, z) = \\|w - z\\|\_2^2$ and some unbounded $\\mathcal{Z} \\subseteq \\mathbb{R}^d$, then one would have infinite $L_{S, U}$. More practically, one may have very large $L_{S, U}$ thanks to some samples $z$ (eg, $z$ that is out of distribution so that the trained weights drastically disagree on them, as in domain generalization scenes).
 2. The paper has provided comprehensive discussion on existing works in trajectory-worst-case generalization bounds and reveals the problem of them. However, beyond these works, mutual-information-based bounds have also tried to solve similar problems of having unbounded IT quantities, which are not cited and discussed in this paper. For example, see [1,2,3] listed below. They essentially inject virtual random (which is also emphasized in this paper) noise and transform unbounded MI terms into bounded and easy-to-estimate MI terms (similar to stability in some sense) and robustness against the noises (similar to complexity and local Lipschitzness). Their drawbacks at least include reliance on local approximation, and assumptions or reliances on testing loss robustness which requires extra samples (like $\tilde{S}$ to estimate Rademacher complexity).
 3. One motivation of the paper is that algorithmic randomness is "paramount to deriving stability-based bounds". However, what specific benefits randomness brings are fully buried in the proof instead of being explained in the main text.
     - I skimmed the proof of Lemma 3.4, where it seems that $U$ does not play important role. Indeed, if one sets $U$ to constant to remove randomness, the proof of course still goes well. It seems the improvement of Lemma 3.4 (eg, the form of empirically relevant Rad + stability) over existing bounds mainly come from some detailed technical improvement instead of exploitation of randomness $U$ (please correct me if wrong).
     - Nevertheless, it seems that randomness $U$ indeed plays important role in reducing $\beta_n$: If randomness is not used, then one has to incorporate randomness by seeing training algorithm instances with seed $U$ (like partial application) as fixed algorithms with different (Foster-style) stability $\beta_{n, U}$, and taking expectation over $U$ (sth like $\\mathbb{E}\_{U} \\mathbb{E}[G_S(\\mathcal{A}\_U(S))] \\le \\mathbb{E}\_{U}[\\dots]$). Then one has to take maximum of $\beta_{n, U}$ over $U$, where some bad $U$ brings extreme looseness. In contrast, with this paper's random-set stability, one essentially replaces this maximum with expectation, which significantly down-weights those bad $U$s.
 4. In experiments, the uniform stability is estimated by taking empirical maximum over hundreds of samples, which is obviously negatively biased.
 5. The motivation and goal of "interplay" experiments are not quite clear to me. Is it validating the theoretical prediction that generalization is approximately proportionate to the complexity with the random-set stability as the factor that decreases as the training set increases? But from Table 1, learning rate, which is altered in the interplay experiments, can drastically change stability, and we should not expect a stable proportionate relation at all.
 6. Minor typos:
     - Lines 035,059,083,167,169,189...: one should use `\citep{}` for these citations
     - Line 866: "index et" -> "index set"?
     - Line 877: "$\omega(S_1, S_2)$" -> "$\omega(\mathcal{W}_{S_1}, S_2)$"?
     - A lot of missing $U$ in the proof of Lemma 3.4?



### References

[1] Gergely Neu, Gintare Karolina Dziugaite, Mahdi Haghifam, and Daniel M. Roy. Information-theoretic generalization bounds for Stochastic Gradient Descent. In Proceedings of Thirty Fourth Conference on Learning Theory, pp. 3526–3545. PMLR, 2021.

[2] Ziqiao Wang and Yongyi Mao. On the generalization of models trained with SGD: Information-theoretic bounds and implications. In International Conference on Learning Representations. ICLR, 2022.

[3] Ze Peng, Jian Zhang, Yisen Wang, Lei Qi, Yinghuan Shi, and Yang Gao. Leveraging Flatness to Improve Information-Theoretic Generalization Bounds for SGD. In International Conference on Learning Representations. ICLR, 2025.

**Questions:**

- Aside from uniform stability that considers the upperbound of hypothesis/loss change, there has been on-average stability that replaces $\sup$ with expectation (eg, [4]), which is usually tighter and easier to estimate. Is it possible to integrate this on-average form into the random-set stability. Will that improve both tightness and estimation (related to my Weakness 4)?




### References

[4] Yunwen Lei and Yiming Ying. Fine-grained analysis of stability and generalization for stochastic gradient descent. In International Conference on Machine Learning, pp. 5809–5819, 2020.

---

> ### Author Response · Authors · 2025-11-21
>
> We thank the reviewer for their positive evaluation of our paper and the pertinent remarks. Below, we address all the concerns raised by the reviewer.
>
> **The auto-satisfaction of local Lipschitzness in Line 315 seems incorrect because the Lipschitzness is not local wrt z. For example, for loss $\ell(w,z) = ||w-z||$ and unbounded $\mathcal{Z} \subset \mathbb{R}^d$ and some unbounded , one would have infinite.** We thank the reviewer for pointing this out; we absolutely agree: our Lipschitz condition is local in the parameter space but global in the data space.We have now made this distinction clear in the revised version, below Assumption 4.1. We would like to highlight the fact that even after taking into account the remark from the reviewer, our assumption is still weaker than the commonly found Lipschitz assumption in this literature, which are often global in both parameter and data spaces (Simsekli et al. (2020); Birdal et al. (2021)).
>
> **For example, see [1,2,3] listed below. They essentially inject virtual random (which is also emphasized in this paper) noise and transform unbounded MI terms into bounded and easy-to-estimate MI terms (similar to stability in some sense) and robustness against the noises (similar to complexity and local Lipschitzness). Their drawbacks at least include reliance on local approximation, and assumptions or reliances on testing loss robustness which requires extra samples (like to estimate Rademacher complexity).**
> We thank the reviewer for the references. Indeed, a popular technique to make MI terms unbounded is to inject Gaussian noise. However, to the best of our knowledge, such a technique has never been proposed for MI terms between random **sets** (which is our case). In particular, the papers mentioned by the reviewer seem to only apply to single-iterate bounds. We believe that extending the noise addition approach to the random set setting would be a promising research direction and we leave it for future work. We now mention these aspects in the future works section, based on this comment from the reviewer.
>
> **I skimmed the proof of Lemma 3.4, where it seems that  does not play important role. Nevertheless, it seems that randomness  indeed plays important role in reducing $\beta_n$.** We thank the reviewer for this comment. Your analysis of our proof of lemma 3.4 is absolutely correct: the randomness does not play a major role in this proof. As it is independent of the other random variables, it is just treated as a constant. However, the randomness is crucial for obtaining low values of the stability parameter, as it is common in the stability literature (Hardt et al. (2016); Zhu et al. (2023)). The reviewer summarizes this correctly: it replaces a maximum by an expectation.
>
> **In experiments, the uniform stability is estimated by taking empirical maximum over hundreds of samples, which is obviously negatively biased.** The reviewer is right to notice that our procedure is estimating a stability parameter that is probably lower than the actual stability parameter.
>
> As these stability parameters are defined as a supremum, we believe that there is no escape to an optimistic estimation of this parameter. We will further emphasize the fact that this estimation is optimistic in the revised version.
>
> **The motivation and goal of "interplay" experiments are not quite clear to me. Is it validating the theoretical prediction that generalization is approximately proportionate to the complexity with the random-set stability as the factor that decreases as the training set increases? But from Table 1, learning rate, which is altered in the interplay experiments, can drastically change stability, and we should not expect a stable proportionate relation at all.** We thank the reviewer for this pertinent remark. Indeed, our experiments are characterized by a strong interdependence of all the relevant quantities (sample size, learning rate, stability, topological complexity, ...), which makes them challenging to conduct. This is a common issue in this entire literature (see for instance the references Andreeva et al. (2024) and  Birdal et al. (2021). In the interplay experiment, we try to validate the dependence of the bound in  $n$ and the topological complexity by letting all the other hyperparameters vary, which has never been done in the prior works. This experiment is especially relevant for our stability analysis and it seems to validate the general form of our bound, but we agree that it does not reveal the exact dependence of the stability parameter on all the other hyperparameters.

---

> > ### Author Response · Authors · 2025-11-21
> >
> > **Typos.** Thank you for pointing that out. We corrected the typos in the revised version. We would like to point out that the subscripts $U$ are not missing in the proof of Lemma 3.4. Indeed, when we write $\mathbb{E}$ without subscript, it is meant that we integrate over all the randomness inside the expectation (including the dataset) this is why the $U$ is not mentioned in the subscript. At line 951, the subscript U indicates that integration is only performed with respect to U (which is possible as U is independent of all the other random variables).
> >
> >  **Aside from uniform stability that considers the upper bound of hypothesis/loss change, there has been on-average stability that replaces  with expectation (eg, [4]), which is usually tighter and easier to estimate. Is it possible to integrate this on-average form into the random-set stability. Will that improve both tightness and estimation (related to my Weakness 4)?**
> > Again, we thank the reviewer for this very pertinent remark. After a quick investigation, it seems that the on-average stability framework could be extended to the random set settings, at the cost of additional assumptions (such as a global Lipschitz assumption), otherwise, the theory would probably be similar to ours. We believe that this remark from the reviewer have interesting empirical consequences, as expectations are typically easier to estimate than supremums. We leave this interesting direction for future works and have now mentioned it in our future works section in the revised version of the paper.

---

### Official Review · Reviewer_aoKU · 2025-11-01

**Soundness:** 2
**Presentation:** 2
**Contribution:** 2
**Rating:** 4
**Confidence:** 3

**Summary:**

This work introduces an interesting class of generalization bounds based on “random set stability”, which is tied to the stability across the parameter trajectory of the algorithm. The aim of these bounds is to bound the worst-case generalization gap across a given window of the trajectory. Crucially, this bound is, unlike information-theoretic relatives, rather straightforward to compute. The authors conclude with experiments on Vision Transformers and GraphSage.

**Strengths:**

1. The paper is clearly structured and articulated.
2. The ability to numerically evaluate these bounds presents a clear advantage over Mutual Information based relatives.

**Weaknesses:**

1. Unless one employs early stopping, the trajectory window at the end of training (assuming near convergence in parameter space) should be essentially a single point, plus tiny fluctuations. I believe that this might lead the presented bounds to be overtly optimistic in near-convergence training settings which is an important and prevalent setting in machine learning. This seems to be consistent to what is written around line 428.
2. In deep learning, Lipschitz constants are typically very large (roughly exponential in the depth). This might further subtract from the achievable accuracy of the presented generalization bounds.
3. I appreciate the inclusion of experiments. However, the results left me a bit confused. It is claimed (and seems to be experimentally supported) that a magnitude scale $\approx \beta_n^{-1/3}$ is a sound choice. This suggests that the bounds in Thm. 4.4 should become better as $n$ increases. However it seems that the bounds in fact become worse for larger sample size (e.g. Pearson correlation $0.28$ for GraphSage with n=10k). Could you please give me your thoughts on this?

**Questions:**

- See weaknesses
- l. 125: unclear what is z_{i,j}

---

> ### Author Response · Authors · 2025-11-21
>
> We thank the reviewer for their positive remarks regarding the structure of our paper and for their thoughtful comments. Below, we address all concerns raised by the reviewer. In light of our responses, we kindly ask the reviewer to consider raising their score.
>
> **I believe that this might lead the presented bounds to be overtly optimistic in near-convergence training settings which is an important and prevalent setting in machine learning. This seems to be consistent to what is written around line 428.**
>
> Regarding the comment that the bound being optimistic, we suspect that there is a confusion about the meaning of the stability over the set. Let us consider a random hypothesis set $W_{S,U}$. The notion of set stability **does not** measure how close the elements of the set $w, w' \in W_{S,U}$ are to each other. Instead it measures how much **the set $W_{S,U}$ itself** changes if we replace the dataset $S$ with another dataset $S'$.
> Hence, even if the set $W_{S,U}$ is collected near convergence, and even if it collapses to a single point, that **does not** mean the set will be stable in our sense (as the stability is measured with respect to the changes in the dataset $S$).
>
> **In deep learning, Lipschitz constants are typically very large (roughly exponential in the depth). This might further subtract from the achievable accuracy of the presented generalization bounds.**
>
> We thank the reviewer for allowing us to clarify this point of our theory.
> Our proof techniques only require (local) Lipschitz continuity over the random set (see Assumption 4.1), which is a less restrictive assumption than global Lipschitz continuity on the entire parameter space. For trajectories obtained by a stochastic optimization algorithm such as ADAM, which are finite hypothesis sets, this assumption is automatically met with a smaller constant in practice.
>
> Further, due to the difficulty of proving generalization bounds for neural networks, Lipschitz continuity has been a common assumption in this literature to simplify the analysis (see  Simsekli et al. (2020); Birdal et al. (2021) and references therein).
>
> **This suggests that the bounds in Thm. 4.4 should become better as $n$ increases. However it seems that the bounds in fact become worse for larger sample size (e.g. Pearson correlation  for GraphSage with $n=10k$). Could you please give me your thoughts on this?**
>
> This is an interesting observation. We believe that the observed behavior might be due to several factors. First, when $n$ increases, the learning task becomes harder and, therefore, it might be more difficult to reach a local minimum. The topological complexities that we report in our experiments are known to be better-behaved near local minimum, so this might explain why our experiments become less noisy and their results less clear when $n$ increases. The reviewer specifically mentions the GraphSage experiments, which are generally more noisy that the image experiments, as it was already reported by (Andreeva et al. 2024). Therefore, this might be another contributing factor to the observed behavior. We added a remark about these aspects in Section 5.1 in the revised version. Moreover, we would like to point out that while our theory predicts that the worst-case generalization error improves as $n$ increases, there is no clear mathematical justification that the Pearson correlation between generalization and topological complexity should improve as $n$ increases.
>
> **l. 125: unclear what is $z_{i,j}$.**  At line 125, we briefly review the setup of Foster et al., 2019. It is based on considering two datasets of size $n$ denoted $S_1$ and $S_{-1}$, with $S_1 := (z_{1,1}, \dots, z_{n, 1})$ and $ S_{(-1)} := (z_{1, -1}, \dots, z_{n, -1})$. Therefore, the notation $z_{i,j}$ simply denotes a datapoint, ie, $z_{i,j} \in \mathcal{Z}$, such that $z_{i,j}$ is an element of the dataset $S_j$ for $j \in \{-1, 1 \}$. We hope that this explaination clarifies this part.

---

> > ### Comment · Reviewer_aoKU · 2025-11-26
> >
> > Thank you for your detailed response!
> >
> > > [...] Instead it measures how much the set $W_{S,U}$ itself changes if we replace the dataset $S$ with another dataset $S'$.
> >
> > Thank you for the clarification, and apologies for getting this wrong in my initial review. However, this now raises another question:
> >
> > If we do collect the points near convergence (where they collapse essentially to a single point), and do the same after fully resampling the whole training set, this will almost certainly lead to a completely different essentially single point, assuming that the model at hand is sufficiently overparametrized. This is certainly the case for standard neural network architectures, where the landscape admits vast amounts of local minima, which may admit essentially the same population risk, but generally err on quite different parts of the sample space. Because of this, typical overparametrized models trained to (near) convergence will not be stable in your sense, and applying your results will yield overly pessimistic generalization bounds (rather than optimistic, as I initially claimed). Would you say that this is an accurate characterization?

---

> > > ### Author Response · Authors · 2025-11-26
> > >
> > > We thank the reviewer for their answer and the opportunity to further clarify our stability condition.
> > > First, we would like to point out that the stability parameter does not correspond to the sensitivity of $W_{S,U}$ to resampling the **whole** training dataset $S$, but its sensitivity when we replace **only one datapoint**, or a **small batch of data points**. For instance, if we have 60.000 samples in our dataset, we imagine that we remove 1 data point from the dataset and replace it with another data point (this is the standard setup in algorithmic stability -- the formal description is written in Assumption 3.1).
> > >
> > > Therefore, for large datasets we can expect that training two identically initialized models on two datasets that only differ by a few datapoints may converge to a similar local minimum.
> > >
> > > Moreover, even if the two networks don't converge to the exact same minimum, we observe that the stability assumption might still be satisfied, as it is a condition on the loss evaluated at these local minima rather than the distance between the local minima (cf. Assumption 3.1 -- and notice that we only look at the difference between the losses, not the iterates).
> > >
> > > Finally, let's make two observations: first, we were able to prove that our random set stability conditions holds for SGD under standard assumptions that are common in this literature.
> > > Second, we would like to point out that all the comments above apply to all the algorithmic stability literature, which is one of the most prominent directions to obtain generalization bounds for stochastic optimization algorithms such as SGD.

---

### Official Review · Reviewer_vF2R · 2025-11-03

**Soundness:** 3
**Presentation:** 1
**Contribution:** 2
**Rating:** 2
**Confidence:** 3

**Summary:**

This paper introduces random set stability, a notion tailored for data-dependent, stochastic hypothesis sets like optimization trajectories. The authors derive an expected worst-case generalization bound that combines this stability parameter (beta_n) with a Rademacher complexity term. This term is computed on the observed random set Cal W_{S,U} using an auxiliary, independent sample. The central claim is that this framework provides computable, trajectory-based bounds without relying on the intractable mutual information (MI) terms that are common in related topological and fractal-based analyses.

**Strengths:**

This paper studies an important and fundamental problem in generalization theory.

**Weaknesses:**

The main weakness of the paper is the writing: it is very dense and the authors don't spend time in giving intuitions etc. Lets consider introduction. The summary of contribution involves discussion on beta_n without precisely defining it. Another example is Definition 3.1, what is G_S'(w)? the definitions need to be self-contained! Lemma 3.2 talks about trajectory-stability. IT hasn't defined clearly. How one can understand the statements?

In this version of the paper, it is very difficult to judge the contribution of this paper.


The other drawback is that this paper mentions that mutual information based bounds are difficult to estimate. For many optimization algorithms, we have results that implies mutual information terms in the bounds can be upper-bounded by simple quantities. For example, look at the following papers:

Negrea, Jeffrey, et al. "Information-theoretic generalization bounds for SGLD via data-dependent estimates." Advances in Neural Information Processing Systems 32 (2019).

Neu, Gergely, et al. "Information-theoretic generalization bounds for stochastic gradient descent." Conference on Learning Theory. PMLR, 2021.

**Questions:**

I have two technical clarification questions.
(1) In Lemma 3.4 you assume ~S_J is independent of S, but in the symmetrization step (and later discussion) it seems you need ~S_J to be independent of the random set W_{S,U}, which depends on both S and the algorithmic randomness U. Should the assumption be ~S_J ⟂ (S,U)? If not, how do you justify symmetrization when the index class is random and data-dependent?

(2) In the projected-SGD corollary, the text summarizes the stability as O(T²/n), but the displayed expression appears to scale like (∑_t η_t)(∑_s η_s L²)/n; how does this reconcile with classical uniform-stability rates O((∑_t η_t)/n)? Could you clarify the precise assumptions (Lipschitz/smoothness/convexity), step-size regime, and whether there’s a typo in the exponents/summary rate?

---

> ### Author Response · Authors · 2025-11-21
>
> Firstly, we would like to thank the reviewer for their constructive comments, especially for pointing out that we address an 'important and fundamental problem.' While we appreciate the reviewer's efforts to improve the manuscript's clarity, we believe that the low score recommendation appears to be based on several misinterpretations. Minor clarifying points have now been addressed in the updated version of the manuscript, and additional justifications of our contributions are made below. We hope that this response addresses the reviewer's concerns and clarifies the significance of our work.
>
> **Unclearness: Authors don't spend time in giving intuitions. The summary of contribution involves discussion on $\beta_n$ without precisely defining it. Another example is Definition 3.1, what is $G_S'(w)$? the definitions need to be self-contained! Lemma 3.2 talks about trajectory-stability. IT hasn't defined clearly.**
>
> * **Intuition and definition of $\beta_n$.** As mentioned in the introduction, we introduce a new stability notion termed *random set stability* governed by a parameter $\beta_n$. Although this stability notion is rigorously defined in the main paper (see Assumption 3.1), we have now referenced the exact definition of $\beta_n$ in the contributions section, along with an intuitive definition to improve readability. To avoid overloading the introduction, an intuitive explanation of $\beta_n$ is provided in the teaser figure, which conceptually clarifies the difference between previous stability notions concerning a single weight vector and our notion concerning random sets.
>
> * **IT has not been defined** We would like to highlight that we introduce the most commonly used information-theoretic (IT) measure, total mutual information, in the introduction (see line 100). Although different studies rely on various other IT quantities, all of these measures essentially attempt to quantify the relationship between the random set $W_{S,U}$ and $S$. The main goal of our paper is to move away from depending on IT terms altogether. Nevertheless, to improve clarity in the revised version, we now direct readers to specific literature containing concrete examples.
>
> * **Definition 3.1**  $G_{S}(w)$ is the generalization gap at a single point $w$, defined in line 48 and used throughout the paper. However, we understand that it might be overlooked; therefore, we now reiterate the full definition in Def. 3.1.
>
> * **Lemma 3.2**: Previously, we incorrectly used 'trajectory stability' instead of 'random set stability' in two places on page 5.
> These have now been replaced with clearly marked changes in the updated manuscript. We thank the reviewer for highlighting this issue, which has improved the clarity of the text.
>
> **For many optimization algorithms, we have results that implies mutual information terms in the bounds can be upper-bounded by simple quantities.** We thank the reviewer for this interesting remark. We would like to point out that the papers cited by the reviewer do not apply to our setting, where we try to prove a worst-case generalization bound over an entire random set, such as the trajectory of a stochastic optimizer. The papers cited by the reviewer concern the mutual information between the dataset and a single vector in $\mathbb{R}^d$ (the output of the algorithm), while the random setting requires to estimate the mutual information between the dataset and a random set that is a **subset** of $\mathbb{R}^d$,  (i.e. $\mathcal{W}_{S,U} \subset \mathbb{R}^d$), which can be easily uncountable. To the best of our knowledge, the only results known to bound the mutual information between random sets are obtained by Dupuis et al. (2024) for noisy SGD and Langevin dynamics, where Gaussian noise is added at every iteration of the algorithm. Extending such analysis to non-noisy (i.e., without additional Gaussian noise) algorithms, such as Adam or SGD (which are the most prominent examples of our framework), is highly nontrivial.
>
> **Should the assumption be $S_J$ independent of $(S,U)$? If not, how do you justify symmetrization when the index class is random and data-dependent?** We thank the reviewer for this comment. We agree that the independence of the dataset $\tilde{S}$ from $U$ should be stated very explicitly. While this is mentioned in the introduction (line 71), we have now clarified this point in the statement of Lemma 3.4 of the revised manuscript.

---

> > ### Author Response · Authors · 2025-11-21
> >
> > **Displayed expression appears to scale like $(\sum_t \eta_t)(\sum_s \eta_s L^2)/n$; how does this reconcile with classical uniform-stability rates $\mathcal{O}((\sum_t \eta_t)/n)$? Could you clarify the precise assumptions (Lipschitz/smoothness/convexity), step-size regime, and whether there’s a typo in the exponents/summary rate?**
> >
> > We suspect there is confusion here that is similar to the previous point. Let us clarify this point in detail.
> > The uniform stability results that the reviewer mentions are again for **single iterate** bounds. Since in Corollary 3.3 we are considering **worst-case** bounds over a trajectory of $T$ iterates, the bounds are hence different by nature.
> >
> > Yet, it is straightforward to reconcile our result with classical single-iterate stability bounds. For a trajectory window of size $T$, our random set stability parameter is indeed of order $T^2 / n$. This is because our framework considers the stability of the entire trajectory, rather than the stability of a single weight vector, as is typically the case in the stability literature.
> >
> > Lemma 3.2 shows that we can upper bound the random set stability parameter by the sum of all the classical single iterate stability parameters, which are in $T / n$. Therefore we obtain $T \times T/n = T^2 / n$.
> >
> >
> > However, if we consider a trajectory consisting of only the last iterate (which is the setting of the bounds that the reviewer mentioned), our reasoning recovers the classical bound that is of order $T / n$.
> >
> > Hence, there is no typo in our result; we recover existing bounds in the single iterate case, and we cover worst-case results over trajectories which cannot be done by standard stability arguments (which yield the $T^2/n$ bounds).
> >
> > Finally, we believe that the assumptions of Corollary 3.3 have already been very clearly stated in the body of the corollary: the loss is taken to be Lipschitz and smooth, with no convexity assumption.
> > The step size is taken as $\eta_k = c/k$, which agrees with the existing literature.

---

### Official Review · Reviewer_UhsB · 2025-11-04

**Soundness:** 3
**Presentation:** 3
**Contribution:** 3
**Rating:** 6
**Confidence:** 2

**Summary:**

This paper proposes a new theoretical framework to explain generalization in deep learning by analyzing the entire training trajectory. To avoid often uncomputable "information-theoretic" (IT) terms found in previous work the authors introduce "random set stability", a measure of how much the worst-case loss along the whole trajectory changes when training data is changed. They show that common algorithms (such as projected SGD) are stable, and then they derive a generalization bound that combines this random set stability parameter and a Rademacher complexity term, recovering some previous bounds as special cases. They claim this offers the first fully computable bound and validate the approach empirically on ViT and GraphSAGE, showing correlations between their bound's components and actual generalization gaps.

**Strengths:**

* Novel stability framework that unifies point-wise points and uniform convergence bounds (by interpolating $J$)
* Computable components

**Weaknesses:**

* Bound has convergence rate of $O(n^{−1/3})$ rather than the usual $O(n^{−1/2})$
* The empirical validation relies on a seemingly very optimistic estimate of the $\beta_n$ parameter
* Empirical results seem to suggest a bound > 1 on a 0-1 loss

**Questions:**

I am afraid I am by no means an expert in this field so my review will have pretty low confidence, but I would love it if the authors could clarify the following:

* Having a slower convergence rate seems like a really big downside. Can the authors please justify why they think this is an acceptable trade-off just to get rid of the information-theoretic terms in the bounds?
* Is the estimation procedure of $\beta_n$ defensible? Assumption 3.1 says that the stability parameter must hold for _any_ value $z \in \mathcal{Z}$, but algorithm 1 simply calculates the maximum over 1000 samples drawn from the training distribution (half seen, half unseen). This seems like a very optimistic estimate. What about points that fall outside of the training distribution? And is 1000 points really enough to estimate this supremum given the high dimensionality of the data? Would it be better to do some sort of hill climbing/optimization to find points for which the difference is large? My concern is that if this estimate is an under-estimate, than all the empirical results will seem much tighter than they actually are, right?

---

> ### Author Response · Authors · 2025-11-21
>
> We thank the reviewer for the positive feedback, including acknowledgment of our  'novel' framework, which yields computationally tractable bounds. In the following, we address the questions raised by the reviewer.
>
> **Convergence rate of $O(n^{-1/3})$.**  The reviewer raises an important theoretical point. Our analysis highlights a key distinction between data-independent and data-dependent learning scenarios. When the hypothesis set $W_{S,U}$ is independent of the dataset $S$, we recover the standard $O(n^{-1/2})$ convergence rate for worst-case bounds (see Corollary 3.6), as well as the usual $O(n^{-1})$ rate associated with classical stability bounds (see Corollary 3.5). In contrast, the hypothesis sets we study are intrinsically data-dependent, as discussed in the introduction, which renders classical tools such as symmetrization and standard stability arguments inapplicable.
>
> Earlier work conducted in the random-set setting relied on techniques that introduce *information-theoretic (IT) terms* in order to decouple the random hypothesis set $W_S$ from the data $S$ (Simsekli et al., 2020; Dupuis et al., 2023; Andreeva et al., 2023), explicit bounds on such IT terms (e.g., the mutual information between $W_{S,U}$ and $S$) have only been established for the Langevin algorithm (Dupuis et al. (2024)). For trajectories produced by algorithms such as SGD or Adam, these IT terms can be unbounded when the data distribution has an uncountable support and there is no additive noise.
>
> Our framework circumvents these limitations by providing worst-case generalization bounds that do not rely on information-theoretic terms. To the best of our knowledge, this yields the first information-theoretic-term-free worst-case generalization guarantees over the trajectories of SGD in non-convex settings, which are provably finite.
>
> *To summarize, we believe that it is better to have a finite and computable bound in $O(n^{-1/3})$ rather than an intractable and potentially infinite bound in $O(n^{-1/2})$.*
>
> **Empirical results on $0-1$ loss.** This phenomenon is a well-known and recurring issue in learning theory [1]. We emphasize that our numerical experiments are designed to assess the order of magnitude by which our bounds deviate from the true values, rather than to obtain exact estimates. The experiments accurately capture the correct order of magnitude of the underlying complexity, which, to the best of our knowledge, is the first demonstration of such behavior for data-dependent worst-case bounds in practically relevant deep-learning settings.
>
> **Is the estimation procedure defensible? Assumption 3.1 says that the stability parameter must hold for any value, but algorithm 1 simply calculates the maximum over 1000 samples drawn from the training distribution (half seen, half unseen).**  This statement is generally true: although the stability parameter can be bounded for certain algorithms (see Lemma 3.2), it is difficult to estimate in the general case.
>
> Due to computational constraints, we estimated the supremum using 1000 samples. We emphasize that half of these samples were taken from an unseen dataset and therefore lie outside the training distribution.
>
> Nonetheless, we agree that this procedure yields an optimistic approximation of the random-set stability parameter. We further stress that the objective of the experiment is not to obtain an exact estimate, but rather to understand the order of magnitude in which the bounds lie. To this end, we also bounded the Rademacher complexity (see line 424). We emphasized the optimistic approximation of the stability parameter in Section 5 in the revised version.
>
> The reviewer's suggestion of using techniques like hill climbing is a very pertinent remark. For classical (single-iterate) stability notions, we believe that it could lead to less optimistic evaluations of the stability parameter. However, as our random set stability assumption (Assumption 3.1) is defined as a Hausdorff-like distance in the data space across two entire trajectories, we believe that directly performing hill climbing on this quantity would be intractable, as it would be very hard to obtain gradients from this quantity, especially given the already high computational cost of these experiments. That being said, we thank the reviewer for this pertinent suggestion, which might be investigated further in future works.
>
> We thank the reviewer again for the valuable remarks, which help to improve the paper further.
>
>  [1] Fantastic Generalization Measures and Where to Find Them, Yiding Jiang* and Behnam Neyshabur* and Hossein Mobahi and Dilip Krishnan and Samy Bengio, ICLR 2020

---

### Official Review · Reviewer_3wS4 · 2025-11-10

**Soundness:** 3
**Presentation:** 3
**Contribution:** 3
**Rating:** 6
**Confidence:** 4

**Summary:**

This paper addresses the challenge of deriving practical worst-case generalization bounds for stochastic optimization algorithms. It introduces random set stability, a novel framework tailored to data-dependent random sets (e.g., optimization trajectories), avoiding intractable mutual information terms in prior works. The framework bounds generalization error via the stability parameter and Rademacher complexity, recovering classical bounds and yielding computable topological bounds.

**Strengths:**

1. This paper introduces the novel concept of random set stability, creatively integrates algorithmic stability with a data-dependent random set framework, successfully avoids intractable mutual information terms in existing methods, and offers a fresh approach to deriving generalization bounds.
2. Through rigorous mathematical deductions, this paper completes theorem proofs and the recovery of classical bounds, while conducting systematic experiments on real datasets using ViT and GraphSage models to validate the effectiveness and tightness of the proposed bounds.
3. This paper provides the first fully computable topological/fractal generalization bounds free of mutual information terms, addresses the core limitation of previous related studies where complexity measures were hard to implement, and advances the application of learning theory in practical algorithms.

**Weaknesses:**

1. This paper only provides expected generalization bounds instead of high-probability bounds, which limits the reliability of the framework in practical scenarios where probabilistic guarantees with strict confidence levels are needed.
2. Estimating the stability parameter requires replacing part of the training samples and retraining, leading to high computational costs in large-sample scenarios without proposing efficient optimization schemes.

**Questions:**

1. Could you please explain the advantage of your defined random set stability in detail compared with others? What if you change $\omega'$ in Assumption 3.1 into $\omega(\mathcal{W}_{S',U},S')$?
2. Is there any possibility of deriving high-probability generalization bounds?
3. Have you explored any efficient estimation methods for the stability parameter $\beta_n$ that avoid complete retraining after sample replacement?

---

> ### Author Response · Authors · 2025-11-21
>
> We thank the reviewer for their positive evaluation of our work, appreciating they recognize the concept of set stability as 'novel', with 'creative' and  'rigorous' proof techniques. We address the reviewer's comments point by point below.
>
>  **Expected generalization bounds instead of high-probability bounds.**
> We thank the reviewer for bringing this point to our attention. Indeed, our bounds are formulated in expectation. This is consistent with a large portion of the stability literature, where many established results — such as those of Hardt et al. (2016) and Zhu et al. (2023), among others — provide bounds in expectation. While obtaining high-probability bounds within our framework is not straightforward, it is certainly an interesting direction for future work.
> We note that even with previously used PAC-Bayes techniques on random sets [Dupuis et al. (2024)] (as discussed in the introduction), reformulating these bounds in expectation would not by itself eliminate the information-theoretic terms.
>
> The removal of these terms is an improvement that is conceptually independent of the choice between expected or high-probability bounds.
>
> **Estimating the stability parameter requires replacing part of the training samples and retraining, resulting in high computational costs in large-sample scenarios.** This observation is generally correct: stability is used to decouple $W_{S,U}$ from the dataset $S$ by quantifying how the random set $W_{S',U}$ changes when $S$ is replaced by a neighboring dataset $S'$. Although we provide results that yield bounds on the stability parameter (see Lemma 3.2), enabling computationally feasible estimations for certain scenarios, we are not aware of any efficient method for computing the stability parameter in full generality without additional assumptions on the algorithm or, more broadly, on the random set itself. To the best of our knowledge, we are also the first who made the effort to numerically evaluate the stability parameters associated to random trajectories in practically relevant empirical settings.
>
>  **Advantage of your defined random set stability compared with others? What if you change $w'$  in Assumption 3.1 into $w(W_{S',U},S')$ ?** Typical stability notions concern single-iterate bounds, such as Algorithmic Stability as introduced by Bousquet (2002) and Hardt et al. (2016) or Uniform Argument Stability by Bassily et al. (2020), which are not applicable to random sets $W_{S,U}$ and therefore not for worst-case bounds. The first step toward addressing this limitation was taken by Foster et al. (2019), who introduced the concept of hypothesis set stability. Our notion is closely related (similar if $U$ is constant) to theirs, but it also accounts for algorithmic randomness, which allows one to obtain worst-case generalization bounds over sets that are dependent on external randomness (e.g., the batch indices in stochastic optimization algorithms).
>
> We also want to emphasize that algorithmic randomness is essential for obtaining bounds on the stability parameter, which makes our assumption more appropriate in the setting of stochastic optimizers, compared to prior art.
>
> Intuitively, the idea can be understood as follows: when training two models on neighboring datasets, we obtain two parameter trajectories. For each weight $\omega \in W_S$ in one trajectory, there exists a corresponding $\omega' \in W_{S'}$, not necessarily at the same iteration, such that their associated losses are close. The measurable function defined $\omega'$ captures this correspondence in the randomized setting.
>
> The change in Assumption 3.1, suggested by the reviewer, would lead to a stronger assumption compared to Assumption 3.1. Indeed, choosing $\omega' = \omega(W_{S'},S')$ means that we require $\omega$ in $W_S$ to be close to the specific selection $\omega'$ in $W_{S'}$ . In comparison, our assumption is weaker as it requires that there exists a point in $W_{S'}$ that is close to $\omega$ (typically the closest), without specifying it.
>
> **Efficient estimation methods for the stability parameter?** To the best of our knowledge, such a stability parameter on a (random) set has not been previously estimated. This is significantly more challenging than estimating the classical (single-iterate) stability parameters, due to the hausdorff-like distance appearing in Assumption 3.1, which requires us to calculate for each $\omega \in W_{S,U}$ the closest $\omega' \in W_{S',U}$, where $S$ and $S'$ differ by a single point.
>
> Consequently, we are not aware of any efficient estimation methods that would avoid complete retraining after sample replacement. Developing such methods could be an interesting direction for future work. For specific cases, the stability parameter can be further bounded, leading to computationally accessible estimates (see Corollary 3.3).
>
> Thank you again for the positive evaluation; we hope that all the concerns you raised have been addressed.

---

### Author Response · Authors · 2025-11-28

We thank the reviewers and AC for their time and feedback, which has helped us improve the manuscript significantly.
We especially appreciate that the reviewers consider our framework **novel** (3wS4, UhsB, PcQB), addressing an **important and fundamental problem** (vF2R, PcQB), supported by **rigorous proofs** (3wS4), with a **clearly structured** presentation (aoKU) that is **easy to follow** (PcQB).

The concerns raised by the reviewer include:  (i) the comparison to single-iterate MI bounds, (ii) the stability parameter estimation, (iii) the validity of our Lipschitz assumptions in deep learning settings, (iv) the convergence rate of our bounds, and (v) minor clarifying details.
We have summarized our responses addressing these points below.
Moreover, we would like to re-emphasize that our main contributions include the **first fully computable topological/fractal generalization bounds** without intractable terms involving mutual information (as also appreciated by reviewers 3wS4 and aoKU).
% We kindly encourage the reviewers to raise any unanswered questions during the remainder of the discussion period.

**(i): Comparison to Single-Iterate MI Bounds (vF2R, PcQB)** We appreciate the reviewers pointing us to works that inject noise to bound MI terms. However, we emphasize that these techniques have been proposed exclusively for **single-iterate bounds**, not worst-case bounds over entire random sets. The mutual information between a dataset and a random *set* (i.e., $W_{S,U} \subset R^d$, which can easily be uncountable) is fundamentally different from MI involving a single vector.
To the best of our knowledge, the only results bounding MI for random sets are obtained for noisy SGD and Langevin dynamics (Dupuis et al., 2024). Extending such analysis to non-noisy algorithms, such as Adam or SGD, is highly nontrivial.
We now mention noise injection approaches in our future works section as a promising research direction.

**(ii): Stability Parameter Estimation (UhsB, PcQB) and Computational Cost (3wS4)** Reviewers UhsB and PCQB observed that our estimation procedure may yield an optimistic approximation of the stability parameter.
We acknowledge this limitation. As the stability parameter is defined as a supremum over all $z \in Z$, any finite-sample approximation will be optimistic. Nevertheless, we stress that (i) the objective of our experiments is to understand the **order of magnitude** in which the bounds lie, not to obtain exact estimates; and (ii) to the best of our knowledge, we are the **first to numerically evaluate stability parameters for random trajectories** in practically relevant empirical settings.

**(iii): Lipschitz Assumptions (PcQB, aoKU)** We clarify that our Lipschitz condition (Assumption 4.1) is **local in the parameter space** but indeed global in the data space.
This is still weaker than the commonly found global Lipschitz assumption in this literature (Simsekli et al., 2020; Birdal et al., 2021).
For finite trajectories produced by stochastic optimizers such as Adam, this assumption is automatically satisfied with smaller constants in practice.
We have clarified this distinction in Assumption 4.1 of the revised manuscript.
Moreover, for trajectories obtained by stochastic optimization algorithms such as Adam, which operate on finite hypothesis sets, this assumption is met with a smaller Lipschitz constant in practice, as opposed to the large Lipschitz constants typically found in deep learning.

---

### Author Response · Authors · 2025-11-28

**(iv): Convergence Rate and Trade-off with IT Terms (UhsB)**

Reviewer UhsB pointed out that our bounds converge in $O(n^{-1/3})$ compared to the classical $O(n^{-1/2})$.
This represents a deliberate and meaningful trade-off centered around our core contribution.
Previous bounds in the random-set setting relied on **information-theoretic (IT) terms** that can be unbounded when the data distribution has uncountable support and no additive noise is present (as in SGD or Adam).
To the best of our knowledge, our framework yields the **first IT-free worst-case generalization guarantees** over random sets in the non-convex settings that are provably finite.

*We emphasize that we believe it is better to have a finite and computable bound in $O(n^{-1/3})$ rather than an intractable and potentially infinite bound in $O(n^{-1/2})$.*

Moreover, we show that when the hypothesis set is data-independent, we recover the standard $O(n^{-1/2})$ rate (Corollary 3.6), and for single-iterate bounds, we recover $O(n^{-1})$ (Corollary 3.5).

**(iv): Clarity (vF2R, aoKU)**

We thank the reviewers for their suggestions, which have improved the clarity of our manuscript. In the revised version:


 -  We have **added intuitive explanations** of $\beta_n$ in the contributions section and referenced the formal definition (Assumption 3.1).
 -  We have **reiterated key definitions** (e.g., $G_S(w)$, IT) where they are used to make statements self-contained.
 -  We corrected terminology inconsistencies (e.g., "trajectory stability" changed to "random set stability") and addressed typos throughout.
 -  We now explicitly state the independence of $\tilde{S}_J$ from both $S$ and $U$ in Lemma 3.4.


In addition to the point-by-point responses, we have incorporated all changes in the revised manuscript, with modifications marked in blue.
We thank the reviewers again for their time and valuable feedback.
We believe that our clarifications regarding the stability parameter estimation, comparison to single-iterate bounds, improved presentation, and strengthened discussion of limitations address the concerns raised.

We have hereby addressed the reviewers’ concerns and believe that, given the opportunity, they would favor positive ratings of our work.

---

### Meta-Review · Area_Chair_XmSp · 2026-01-04

**Summary:**

This paper proposes new computable generalization bounds via random set stability. Three reviewers (3wS4, UhsB, and PcQB) recommend acceptance, while two (vF2R and aoKU) recommend rejection. The main concerns include presentation quality, comparisons with existing computable MI bounds, how the stability parameters are estimated in practice, the role and plausibility of Lipschitz assumptions, the slow convergence rate of the bound, and whether the expectation bound has a high-probability analogue.

Several points seem to have been addressed adequately in the rebuttal or revision (e.g., improvements in presentation and additional clarification on how the stability parameters can be estimated). That said, I believe some concerns remain.

Most notably, the slow convergence rate is still worrisome. While the authors argue that computational tractability can be preferable to sharpness, I do not find this fully convincing. In learning theory, the tightness of a bound is highly related to whether it is sufficiently informative about generalization. If a bound is too loose, it becomes unclear what we can trust it to explain (no matter computable or not). This also raises another question, that is, beyond computability, what new insight about generalization does the proposed bound provide? In my view, computability alone is not the end goal, and the bound should ideally support understanding or actionable guidance. From that perspective, faster rates (or at least demonstrating nontrivial tightness in meaningful settings) seems important.

Moreover, although the authors emphasize that certain computable MI-based bounds are single-iterate in nature, those bounds have been practically useful because they connect generalization to interpretable quantities along the training trajectory, e.g., gradient norms/variance and solution flatness, which can, in turn, motivate algorithm design. If one can obtain bounds that are both computable and insightful at the single-iterate level, what is the practical motivation for focusing on computable worst-case bounds? As a side point, even from a purely theoretical perspective, it would be helpful to clarify how the paper's results relate to recent work questioning whether uniformly tight generalization measures can exist at all [1].

Overall, I appreciate the progress made in the revision, but I hope these remaining concerns, especially the informativeness of the bound in view of its rate and the practical motivation for worst-case computable bounds, can be more fully addressed in a further revision.

[1] Michael Gastpar et al., "Fantastic Generalization Measures are Nowhere to be Found." ICLR 2024.

**Reviewer Concerns:**

Reviewer 3wS4's concerns about the lack of a high-probability guarantee, along with several clarification requests (e.g., the computational cost of estimating the stability parameter), have been addressed. In particular, the authors explain that expectation-type generalization bounds are common in the literature and that extending random set stability to high-probability bounds is nontrivial. I find this response reasonable.

Reviewer UhsB also asks about how the stability parameter is estimated, and the authors' rebuttal largely resolves this point. However, I do not think the reviewer's remaining concerns, especially the slow convergence rate and the numerical looseness of the bound, are fully addressed, for the reasons discussed above.

Reviewer vF2R raises issues about writing quality and comparisons with MI-based bounds. These may have been partially mitigated (the presentation has improved), but the comparison to existing computable MI bounds remains insufficiently convincing.

Reviewer aoKU mainly requests clarifications and expresses concern about the Lipschitz assumptions. These points seem to be mostly addressed in the rebuttal.

Reviewer PcQB similarly questions the Lipschitz assumptions, the comparison with MI-based bounds, and several technical details. Most clarification points are addressed, but I think that the comparison to MI-based bounds remains an unresolved issue, as noted above.

**Reviewer Scores:**

Reviewer 3wS4 and Reviewer UhsB are likely to maintain their overall positive evaluations, although some of Reviewer UhsB's concerns (particularly regarding the slow rate and numerical looseness) may remain only partially addressed.

Reviewer vF2R is, in my view, unlikely to increase the score. Although the revision improves presentation quality, this reviewer seems highly critical of the writing and may be looking for more substantial restructuring. Moreover, the rebuttal on the comparison with MI-based bounds may not be convincing to this reviewer.

Reviewer aoKU may increase the score, since the main concerns seem to have been addressed.

It is less clear whether Reviewer PcQB will update the score. While several technical clarifications were resolved, this reviewer may remain unsatisfied with the comparison to MI-based bounds.

Overall, the paper's final score may be quite borderline, and it is not obvious that any reviewer will strongly champion acceptance.

---

### Decision · Program_Chairs · 2026-01-26

Reject